# Finetune Once: Decoupling General & Domain Learning with Dynamic Anchor Annealing

## Abstract

Large language models (LLMs) fine-tuning shows excellent implications. However, vanilla fine-tuning methods often require intricate data mixture and repeated experiments for optimal generalization. To address these challenges and streamline the training process, we propose an efficient and universal solution, Dynamic Anchor Annealing (DAA). We obtain a global gradient through zero-learning-rate training on general data, which is subsequently employed for gradient anchor and dynamic training step correction during domain training. In conjunction with annealing learning, we end up establishing a fine-tuning pipeline that relies solely on domain data without collapse. By evaluating both general and domain-specific performance across multiple tasks on several popular base models, DAA achieves an average improvement of 5.8% in joint performance over vanilla fine-tuning. Furthermore, since general data is no longer involved in annealing, repeated experiments led by data mixture are also eliminated. According to our tests, the DAA method can reduce GPU hours by 91.0% compared to the vanilla method.

## 1 Introduction

Large Language Models (LLMs) show significant promise in various applications due to their ability to understand and generate human-like text. Fine-Tuning (FT) LLMs on domain-specific tasks has become a common approach to enhance their performance in targeted applications Yang et al. (2023); Zhou et al. (2024); Chen et al. (2024); Huang et al. (2023). However, empirical evidence suggests that fine-tuned LLMs frequently demonstrate significant degradation of their original performance Chen et al. (2020); Luo et al. (2025); Lin et al. (2023); Korbak et al. (2022). Therefore, mitigating catastrophic forgetting in the fine-tuning process has emerged as a crucial research focus for LLMs (Table 1, row 1).

Data Mixture (DM) strategy was the basic and vanilla solution Wen et al. (2023); Wu et al. (2023); Zhang et al. (2024a); Wu et al. (2024); Held et al. (2025) to solve catastrophic problem. It combines general and domain-specific data in fine-tuning datasets to mitigate forgetting of general capabilities. Due to the coupling between data from different domains, each fine-tuning requires repeated experimentation to adjust the data mixture in order to achieve satisfactory performance (Table 1, row 2). As shown in Figure 1, the effectiveness of DM heavily depends on the mixing ratio, necessitating extensive empirical validation to determine optimal proportions for each domain. Alternative approaches, such as Low-Rank Adaptation (LoRA) Hu et al. (2021); Yang et al. (2023); Cui et al. (2023), have demonstrated some success in preserving general capabilities, yet they face inherent limitations in achieving peak domain-specific performance (Table 1, row 3). This ad-hoc process of data mixing is not only computationally prohibitive but also lacks scalability, as the optimal ratio for one domain rarely transfers to another. Consequently, an ideal fine-tuning framework must decouple domain adaptation from the costly cycle of data mixture experiments, while still effectively balancing specialization with the preservation of general knowledge.

To address the above challenges, we propose **Dynamic Anchor Annealing (DAA)**, a streamline fine-tuning framework that eliminates the requirements for data mixture and repeated experiments. First, to effectively isolating the contributions of general-domain and domain-specific data, we propose **Global Gradient Anchoring (GGA)**. Here, "Anchoring" refers to augmenting the domain-specific gradient with a pre-computed global one. This method initially estimates the global gradient in the general domain through zero-learning-rate learning. During fine-tuning, the global gradient is

Figure 1: Comparison between vanilla and DAA. [*] is the part that users need to perform in SFT.

Table 1: Comparison of different methods. Repeated Exps indicates that the method requires hyper-parameter tuning or recipe adjustment for data mixture ratios to achieve optimal results. Collapse means losing generalization ability.

| Method | No Data Mixture | No Repeated Exps | Reduce Cost | No Collapse | SOTA |
|---|---|---|---|---|---|
| Direct FT | ✔ | ✔ | ✔ | ✗ | ✗ |
| Vanilla FT | ✗ | ✗ | ✗ | ✔ | ✗ |
| LoRA-like FT | ✔ | ✗ | ✔ | ✔ | ✗ |
| **DAA (Ours)** | ✔ | ✔ | ✔ | ✔ | ✔ |

used as guidance, combined with annealing learning, to mitigate catastrophic forgetting. Second, to achieve global optimal performance in specifics, we propose a domain similarity-guided **Dynamic Correction (DC)** strategy. This adaptive parameter update strategy modulates the optimization steps based on the gradient similarity between specific and general domains. It is important to emphasize that the fundamental goal of DAA extends beyond the reduction of fine-tuning expenses to specifically address the mitigation of catastrophic forgetting. The efficiency of this method emerges as a significant inherent benefit of its design, as it delivers comparable or superior performance while avoiding the substantial costs associated with data mixture search. As demonstrated in Table 1, DAA achieves superior performance compared to conventional fine-tuning approaches, while significantly reducing workload by eliminating the need for data mixing and repeated experiments. Our contributions are summarized as follows:

- We explore the impact of data mixture on both fine-tuning performance and workload, and propose new fine-tuning schemes.
- We propose DAA, a novel training framework designed to efficiently fine-tuning by gradient-based domain decoupling and similarity-guided adaptation.
- We conduct empirical evaluations across various tasks, demonstrating that our method effectively balances domain-specific performance while maintaining general capabilities with low workload.

## 2 MOTIVATION

### 2.1 RELATED WORK

Recent work on fine-tuning Xie et al. (2023); Zhang et al. (2023b); Bao et al. (2023); Yue et al. (2023); Chen et al. (2024); Zhou et al. (2024); Yang et al. (2023); Cui et al. (2023) including direct fine-tuning Xie et al. (2023); Zhu et al. (2024), vanilla fine-tuning Bao et al. (2023); Yue et al. (2023); Chen et al. (2024); Deng et al. (2023) and LoRA Yang et al. (2023); Chen et al. (2023); Cui et al. (2023) seeks to mitigate catastrophic forgetting while controlling cost. As shown in Table 1, direct fine-tuning is inexpensive yet the general performance of LLMs can collapse. Vanilla fine tuning employs data mixtures to suppress forgetting and often preserves general capability, although

the cost rises sharply. LoRA is effective in reducing both forgetting and cost, but performance in unfamiliar specific domains remains below that of full-parameter fine-tuning.

Among these options, vanilla fine-tuning with data mixture offers the best balance between general and domain-specific performance, yet its experimental cost is substantial. Mixing ratios tend to be domain- dependent and therefore require repeated experimentation for each target domain Wen et al. (2023). In addition, when the ratio is swept from $1:1$ through $1:N$, the total volume of processed data scales as $\sum_{n=1}^{N}(1+n) = O(N^2)$ times the size of the specific domain set, which becomes prohibitive as the domain dataset grows. This computational inefficiency motivates more efficient and more generalizable fine-tuning methodology.

Recent studies Wu et al. (2024); Dong et al. (2024) investigate the optimization of data mixtures. For instance, Mixture-of-Skills Wu et al. (2024) utilizes Reinforcement Learning (RL) to optimize data utilization ratios. While DAA eliminates the need for online mixture search, it is orthogonal to data selection strategies. Future work could integrate RL-based data curriculum to further enhance the quality of the Global Gradient Anchor. Additionally, Liang et al. (2025) employs a sample-level Evolving Interaction-guided Curriculum for multi-domain coordination, DAA adopts a domain-level strategy through the Global Gradient Anchor. This approach abstracts the general prior into a stable optimization direction to prioritize computational efficiency and scalability over complex sample dynamics, which facilitates a streamlined "Finetune Once" workflow. Furthermore, gradient surgery methods such as PCGrad Yu et al. (2020) and GradNorm Chen et al. (2018) manage gradient conflicts in multi-task learning by storing individual task gradients, yet this approach incurs prohibitive memory costs for LLMs. In contrast, DAA employs a memory-efficient and pre-computed global anchor to specifically mitigate catastrophic forgetting.

Another related area of research is Continual Learning (CL). CL is defined as a model learning from a dynamic data distribution Wen et al. (2023). Our setting can be viewed as single task continual learning in which, after adapting a pretrained model to one instruction task, we aim to mitigate degradation of its general capabilities.

Besides, it's important to differentiate the optimization challenges in DAA from those in Multi-Task Learning (MTL). Prior MTL works often focus on **imbalanced domains**, where high-resource tasks dominate the gradient direction at the expense of low-resource tasks Sener & Koltun (2018), and **cross-domain transfer**, which aims to leverage knowledge from a source domain to improve a distinct target domain Liu et al. (2019). In contrast, DAA targets Finetuning to Target Domain , where the optimization landscape is defined by the tension between the General Prior and the Target Domain. 1) Handling Imbalance: Unlike MTL, where multiple specific domains compete for model capacity, the imbalance in our setting exists between the massive pre-training distribution and the smaller fine-tuning dataset. DAA explicitly regulates this imbalance through the Global Gradient Anchor and the decay coefficient $\gamma_t$, ensuring that the general prior is not overwhelmed by the specific domain, regardless of the domain's data scale. 2) Nature of Transfer: DAA focuses on General-to-Specific retention rather than Specific-to-Specific transfer. The Global Gradient serves as a regularization term that enforces the preservation of general reasoning structures while allowing the model to adapt to the target distribution. Thus, the negative transfer or interference often seen between disjoint domains in MTL is mitigated by anchoring optimization to the robust general manifold.

## 2.2 Role of Gradient in fine-tuning

During stochastic optimization, gradient variance strongly affects convergence and generalization Gurbuzbalaban et al. (2021). High variance slows convergence and complicates optimization Agarwal et al. (2022); Xia et al. (2024), which can hinder domain adaptation. We provide qualitative and quantitative analyses of fine-tuning across general and specific domains and expose drawbacks of multi domain optimization.

First, we qualitatively analyze the differences in convergence trajectory between general and specific domain by visualizing the loss landscape. Following the methodology in Lucas et al. (2021), we interpolate between the weights $\theta_0$ of Qwen3-1.7B Yang et al. (2025) base model and the fully fine-tuned weights $\theta_D$, constructing a two-dimensional slice of the loss landscape. To ensure independence, we apply orthogonalization to the interpolation direction.

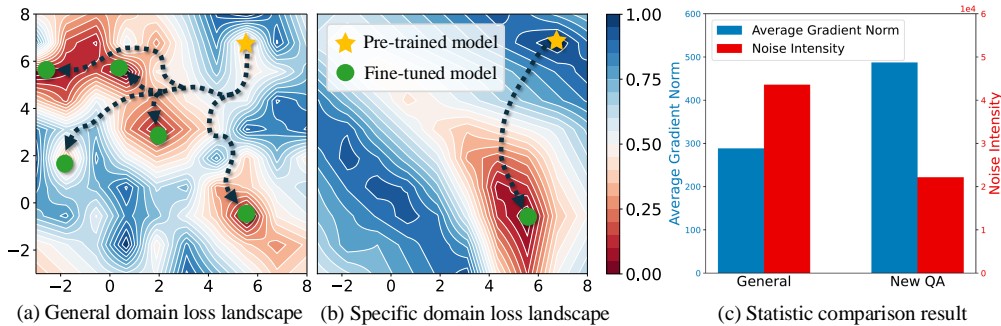

(a) General domain loss landscape    (b) Specific domain loss landscape    (c) Statistic comparison result

Figure 2: Qualitative and quantitative analysis result of general and specific domain.

Figure 2(a) and 2(b) show that the general domain has many local optima and a tortuous path details in appendix B, while the specific domain news QA details in Section 4.2 shows fewer optima and a more stable trajectory. General domain training can therefore constrain domain specific fine-tuning.

Second, we follow stochastic optimization methodology Ghadimi & Lan (2013) to compare average gradient norms and noise scale across the two domains. As shown in Figure 2(c), the general domain has nearly twice the noise scale of the specific domain. We randomly sample 1,000 instances from each domain. On the full general domain the gap may be larger. We therefore attempt to freeze general domain gradients to limit their impact on training.

In multi domain optimization, conflicts between domain gradients degrade efficiency Yu et al. (2020); Hadsell et al. (2020); Liu et al. (2021). Prior work reduces negative interactions by removing projection components between domain gradients Yu et al. (2020) or by automatic gradient balancing Liu et al. (2021). This motivates a balancing mechanism between specific and general domain gradients that preserves generalization while learning specific domain distributions.

## 3 METHOD

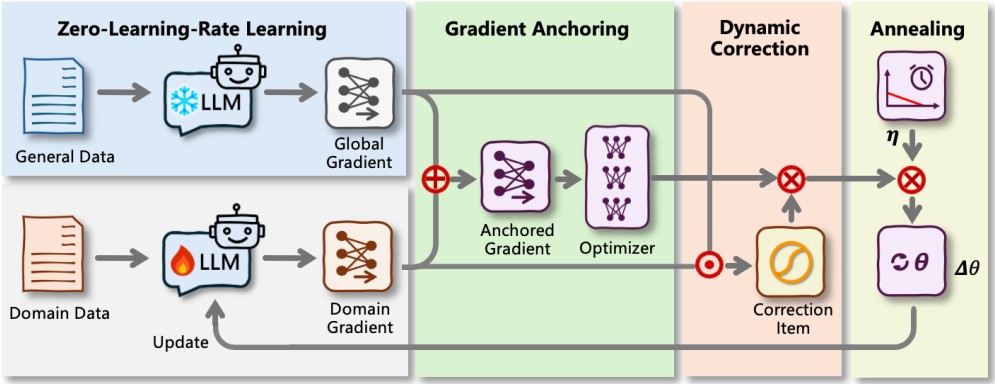

Figure 3: Overview of Dynamic Anchor Annealing. Our approach consists of two stages. In the first stage, global gradient is estimated in the general domain through zero-learning-rate learning, which serves as an independent preprocessing stage. In the second stage, the fine-tuning step, global gradient anchors the specific gradient to preserve general capability, while the similarity between global and specific gradients adaptively determines the parameter update magnitude. The learning rate with annealing strategy suppresses degradation.

In this section, we formally introduce the Dynamic Anchor Annealing (DAA) illustrated in Figure 3, which is based on annealing learning. It is worth to note that DAA is a full-parameter fine-tuning framework in which decoupling refers to the optimization-level separation of general and domain-specific signals within the gradient space rather than the division of model components. Initially,

the global gradient is independently estimated in the general domain through zero-learning-rate learning. During the fine-tuning stage, DAA anchors the gradient to preserve the general capability. Subsequently, the similarity between the global gradient and specific gradient adaptively selects the magnitude of parameter update. Finally, the learning rate with the annealing strategy suppresses degradation effectively.

## 3.1 GLOBAL GRADIENT ANCHORING

The global gradient serves as a stable optimization anchor. Our design is motivated by two key factors: 1) Variance Reduction: $\hat{g}_G$ acts as a low-variance regularizer for the high-variance domain gradient $g_{D,t}$, preventing severe oscillations (see Eqn. 4). 2) Trajectory Smoothing: As shown in Figure 2(a), $\hat{g}_G$ points towards regions of good generalization, helping the optimizer bypass poor local optima in the general domain landscape. The term "Anchoring" is metaphorical. We anchor or augment the domain-specific gradient at each step with this pre-computed stable anchor. This approach is distinct from traditional Gradient Boosting Machines.

In the joint learning of general and specific domains, the gradient is a weighted sum of the general gradient and the specific gradient, with weights determined by the data mixing ratio $\lambda$, that is

$$g_{M,t} = \lambda g_{G,t} + (1 - \lambda) g_{D,t}. \tag{1}$$

We define $\hat{g}_G$ as a fixed estimator of $g_{G,t}$ in joint training to diminish the volatility of the combined gradient.

$$g_{B,t} = \gamma_t \hat{g}_G + (1 - \gamma_t) g_{D,t}, \tag{2}$$

where $\gamma_t$ is the anchoring magnitude. The expectation and variance of $g_{B,t}$ are given by

$$\mathbb{E}[g_{B,t}] = \gamma_t \hat{g}_G + (1 - \gamma_t)\mathbb{E}[g_{D,t}], \tag{3}$$

$$\mathbb{E}[\|g_{B,t} - \mathbb{E}[g_{B,t}]\|^2] = (1 - \gamma_t)^2 \mathbb{E}[\|g_{D,t} - \mathbb{E}[g_{D,t}]\|^2]. \tag{4}$$

By fixed $\hat{g}_G$, we can significantly mitigate the randomness of parameter update while maintaining the regularization effect on optimization.

To estimate the global gradient $\hat{g}_G$, according to the derivation of Adam Kingma & Ba (2017), when the exponential decay rate for the 1$^{\text{st}}$ momentum estimates $\beta_1 \to 1$, the momentum $m_{G,t}$ approximates the expectation of the gradient.

$$\hat{g}_G = \mathbb{E}[g_{G,i}] = s^{-1} \sum_{i=1}^{s} g_{G,i} = \lim_{\beta_1 \to 1} m_{G,s}. \tag{5}$$

Therefore, we trained the LLM on the general domain with a learning rate of 0 and a decay rate $\beta_1 \to 1$, then stored the final momentum after $s$ training steps. Notably, $\hat{g}_G$ can be applied across all domains, rather than being obtained per domain. The acquisition of $\hat{g}_G$ is consistent with the findings of LIMA Zhou et al. (2023) that prioritize diversity and quality over sheer quantity. As the general dataset attains sufficient diversity, the direction of $\hat{g}_G$ stabilizes, which implies that massive datasets are not strictly required for robust anchoring.

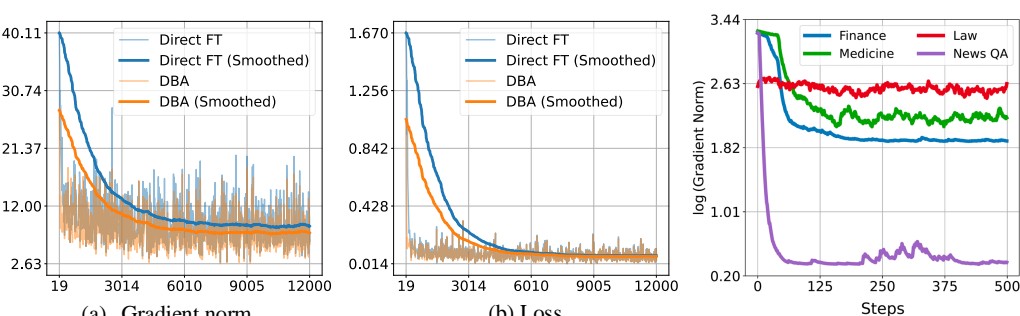

(a)  Gradient norm           (b)  Loss

Figure 4: Comparative results of global gradient boosted learning.      Figure 5: Norm of gradient projection.

We refer to this method as Global Gradient Anchoring (GGA). The stored $m_G$ from the general domain supplies a stable guidance signal during specific domain optimization and steers updates toward a joint optimum. As shown in Figure 4, GGA markedly reduces gradient norm and loss, especially early in fine-tuning, which indicates improved training stability on the specific domain with preservation of general capability.

The anchoring magnitude can be adapted over training. We set $\gamma_t = k_0\left(1 - \frac{t}{T}\right)$, so that the model increasingly emphasizes the specific domain near the end of fine tuning. Accounting for exponential averaging in Adam Kingma & Ba (2017), the exponential average of $\gamma_t m_G$ yields an effective coefficient $\alpha$:

$$\alpha_t = k_0\left(1 - \frac{t}{T}\right) + \frac{k_0\beta_1\left(1 - t\beta_1^{t-1} + (t-1)\beta_1^t\right)}{T(1-\beta_1)(1-\beta_1^t)}. \tag{6}$$

The nonlinear term is monotonically increasing and is bounded by $\frac{k_0\beta_1\left(1-t\beta_1^{t-1}+(t-1)\beta_1^t\right)}{T(1-\beta_1)(1-\beta_1^t)}$. Since the cumulative contribution of the global gradient should be comparable across domains, it suffices to choose $k_0$ inversely proportional to $T$. With $T \gg k_0$, the nonlinear term becomes negligible and we use the approximation $\alpha_t \approx k_0\left(1 - \frac{t}{T}\right)$. This schedule is simple to implement and robust. Hyperparameter details and sensitivity analyses are provided in the Appendix A.

For deployment efficiency, storing $m_G$ in 32 bit precision is memory intensive. We therefore apply singular value decomposition to $m_G$ and retain a rank $r = 512$ approximation. During training, we reconstruct the low rank estimate of the global gradient and add it to each step. This saves memory and emphasizes the most informative components of the global signal.

## 3.2 DYNAMIC CORRECTION

Pre-trained language models have demonstrated remarkable capabilities by incorporating data from diverse domains during pre-training. However, these models often struggle with domains that are either private or temporally distinct from the pre-training distribution, necessitating extensive experiments for optimal performance. Applying uniform strategies across domains with varying degrees of familiarity can lead to suboptimal outcomes. To address this challenge, we propose a Dynamic Correction (DC) mechanism that modulates the magnitude of parameter update based on gradient similarity. The core intuition is to adjust the learning step based on the interference strength $s_t$. When $s_t \approx 0$, gradients are orthogonal (low interference), allowing larger steps. When $s_t \approx 1$, gradients are correlated (high interference), requiring smaller steps to preserve general knowledge.

To quantify the alignment between general domain and specific domain, we introduce a gradient similarity metric based on the L2 norm of the normalized projection of specific gradient $g_{D,t}$ onto the estimation of the global gradient $m_G$:

$$s_t = \frac{||g_{D,t} \cdot \hat{g}_G||}{||g_{D,t}|| \cdot ||\hat{g}_G||}, \tag{7}$$

where $g_{D,t}$ denotes gradients for the specific domain at time step $t$, and $\hat{g}_G$ represents global gradient of the general domain. Our empirical analysis encompasses both familiar domains (finance Yang et al. (2023), medicine Wang et al. (2024), and law Fei et al. (2023)) and a temporally restrictive domain (news QA, details shown in section 4.2). As shown in Figure 5, each domain maintains a characteristic similarity range with the general domain. Notably, familiar domains exhibit similar magnitude, while the unfamiliar domain demonstrates significantly lower similarity values, differing by more than an order of magnitude.

Leveraging this similarity measurement, we introduce a dynamic correction coefficient:

$$c_t = s_t + c_0, \tag{8}$$

where $c_0$ represents a base coefficient that prevents excessive parameter updates and potential overfitting when similarity values are minimal. We set $c_0 = 0.01$ in practice. The resulting parameter update rule incorporating the dynamic correction is:

$$\Delta\theta = -\eta\frac{\hat{m}_t}{\sqrt{c_t\hat{v}_t} + \varepsilon}. \tag{9}$$

## 3.3 ANNEALING LEARNING

In the training of MiniCPM Hu et al. (2024) and Llama3 Grattafiori et al. (2024), Annealing Learning (AL) is applied at the final stage of pre-training. Using learning rate with minimal initialization and decay strategy, LLMs can learn downstream task knowledge from high-quality domain data without forgetting. Suppose the conventional initialization of learning rate is $\eta_0$, and $\eta_0^{\mathrm{a}}$ for annealing, the parameter updates for both schemes are:

$$\Delta\theta_t = -\eta_0\left(1 - \frac{t}{T}\right)\frac{\hat{m}_t}{\sqrt{\hat{v}_t} + \varepsilon}, \quad \Delta\theta_t^{\mathrm{a}} = -\eta_0^{\mathrm{a}}\left(1 - \frac{t}{T}\right)\frac{\hat{m}_t}{\sqrt{\hat{v}_t} + \varepsilon}. \tag{10}$$

Via comparative analysis of Eq. 10, we can measure the influence of annealing on the parameter updates:

$$\Delta\theta_t - \Delta\theta_t^{\mathrm{a}} = -(\eta_0 - \eta_0^{\mathrm{a}})\left(1 - \frac{t}{T}\right)\frac{\hat{m}_t}{\sqrt{\hat{v}_t} + \varepsilon}. \tag{11}$$

As shown in Eq. 11, annealing suppresses the learning of specific domains. Smaller parameter updates thus reduce the risk of catastrophic forgetting. Therefore, we use the annealing learning scheme in DAA. And we set $\eta_0^{\mathrm{a}} = 1e^{-7}$ in the our experiment.

## 3.4 SUMMARY

After integrating the above learning strategies, we obtain the complete parameter update of DAA:

$$\Delta\theta_t^{\mathrm{DAA}} = -\eta_0^{\mathrm{a}}\left(1 - \frac{t}{T}\right)\frac{\hat{m}_{B,t}}{\sqrt{c_t\hat{v}_{B,t}} + \varepsilon}. \tag{12}$$

The integration of GGA, DC and AL facilitates the adaptation to specific domains while mitigating forgetting.

## 4 EXPERIMENT

### 4.1 EXPERIMENT SETTINGS

This study evaluates the effectiveness of DAA across diverse vertical domains in both English and Chinese contexts, including finance, medicine, and law. The general-domain data used in our experiments comprises Chinese and English corpora covering multiple tasks. Its detailed composition can be found in Appendix B, Table 6. The evaluation utilizes multiple datasets: FinGPT Yang et al. (2023), CMB Wang et al. (2024) and Fuzi-Mingcha Deng et al. (2023). To avoid the potential contamination or overfitting of evaluation benchmarks during pre-training as new and improved LLMs are developed Schaeffer (2023); Jain et al. (2024); Zhang et al. (2024b), we constructed a temporal out-of-distribution (OOD) evaluation benchmark named News QA (details in Section 4.2).

For comparative analysis, we selected several representative fine-tuning methods. In addition to direct fine-tuning and vanilla fine-tuning, we also compared the performance of LoRAHu et al. (2021), DoRALiu et al. (2024), GaloreZhao et al. (2024), and our proposed DAA across diverse vertical domains. Especially, for vanilla fine-tuning, we followed Wen et al. (2023) and combined our vertical domain fine-tuning experience to choose three distinct data mixture ratios (specific data : general data = 1:1, 1:3, 1:5). We ensured that the vanilla fine-tuning results presented in the experimental tables all represent the optimal performance in the specific domain.

In addition, we have validated the effectiveness of the proposed method across multiple foundational models, including Llama3.1-8B Grattafiori et al. (2024) , Phi4-14B Abdin et al. (2024), and Qwen3-8B Yang et al. (2025).

### 4.2 NEWS QA BENCHMARK

We constructed a benchmark comprising QA pairs extracted from news articles. As shown in Table 2, the dataset contains 30,613 news titles across three categories (Politics, Economics, and Culture), with corresponding true/false questions designed to evaluate factual verification capabilities. The

task requires binary responses ("true" or "false") for each statement. To ensure minimal overlap with foundational models' pre-training corpus Yang et al. (2024); Grattafiori et al. (2024); Abdin et al. (2024), we specifically selected news articles published after December 2024. As shown in Table 2, row 3, all the foundational models exhibits limited factual verification capabilities , achieving only 31.06% average accuracy.

Table 2: Details of news QA benchmark. The first two rows show the data distribution. The third row presents performance ($S_D$) of Qwen3-8B Yang et al. (2025) on each category.

| Split | Politics | Econ | Culture | Total |
|---|---|---|---|---|
| Train set | 9823 | 10120 | 8670 | 28613 |
| Test set | 700 | 700 | 600 | 2000 |
| $S_D$ | 29.05 | 30.70 | 33.43 | 31.06 |

### 4.3 METRICS

To evaluate the general performance of the models, we selected four benchmarks commonly used across all LLMs: MMLU Hendrycks et al. (2021a), MMLU-Pro, GSM8K Cobbe et al. (2021), MATH Hendrycks et al. (2021b) and M3Exam Zhang et al. (2023a). MMLU tests general knowledge across multiple subjects, CMMLU focus on Chinese-specific knowledge and reasoning, while GSM8K and MATH tests mathematical problem-solving skills. All raw scores for the evaluated benchmarks were within the interval of [0, 1]. Crucially, no external normalization procedures were applied during the computation of $S_D$, $S_G$, and $S$. To evaluate various vertical performance of the models, we selected suitable public benchmarks for evaluation. For the financial domain, we utilized the weighted F1 score average across the English FPB Malo et al. (2013), FiQA Maia et al. (2018), TFNS Zer (2024), and NWGI Yang (2024) financial sentiment analysis test sets as the metric for this domain. For the medical domain, the accuracy score from the Chinese CMB-Exam Wang et al. (2024) test set served as the domain-specific metric. For the legal domain, we employed Chinese LawBench Fei et al. (2023) for a comprehensive evaluation. Our overall metric design is structured as follows:

$$S_G = \text{Mean}(\{S_x \mid x \in \mathcal{X}\}), \tag{13}$$

$$S = \text{HarmonicMean}(S_D, S_G), \tag{14}$$

where $\mathcal{X} = \{\text{MMLU, MMLU-Pro, GSM8k, MATH, M3Exam}\}$. $S_D$ is the score of the model's vertical domain performance, $S_G$ is the average score of the model's general performance. $S$ is the harmonic mean of $S_D$ and $\Delta S_G$, meaning the model scores high only if both are large. If fine-tuning boosts domain performance but reduces general capability significantly, the score nears 0. Conversely, if it enhances domain performance while preserving general capability, the score approaches 1.

### 4.4 COST ANALYSIS

To demonstrate the efficiency of our approach, we compared GPU hours across fine-tuning methods using 16 Nvidia A100 GPUs. As shown in Table 3, DAA requires $T_{\text{DAA}} \approx 4.2$ GPU-hours per domain, similar to direct full-tuning ($T_{\text{Direct}} \approx 4.1$ GPU-hours) but without its drop in general-task performance. More importantly, DAA reduces training costs by over 90% compared to vanilla fine-tuning ($T_{\text{Vanilla}} \approx 46.7$ GPU-hours) while achieving notable gains in vertical ($S_D$) and general ($S_G$) scores.

While LoRA ($T_{\text{LoRA}} \approx 3.0$ GPU-hours) and DoRA ($T_{\text{DoRA}} \approx 3.1$ GPU-hours) are faster, DAA consistently outperforms them and Galore ($T_{\text{Galore}} \approx 5.1$ GPU-hours) in harmonic-mean score $S$, justifying the modest additional GPU time with superior task performance.

### 4.5 MAIN RESULTS

Table 3 demonstrates that our DAA method consistently achieves the best balance between vertical domain ability and general performance retention, as measured by the harmonic mean $S$, across four domains and three base models.

Table 3: Performance metrics across different domains and models. $T$ is the GPU hours. $S_D$ is the score of the model's vertical domain performance, $S_G$ is the average score of the model's general performance change. $S$ is the harmonic mean of $S_D$ and $S_G$.

| Domain | Method | Llama3.1 | | | | Phi4 | | | | Qwen3 | | | |
|---|---|---|---|---|---|---|---|---|---|---|---|---|---|
| | | $T\downarrow$ | $S_D\uparrow$ | $S_G\uparrow$ | $S\uparrow$ | $T\downarrow$ | $S_D\uparrow$ | $S_G\uparrow$ | $S\uparrow$ | $T\downarrow$ | $S_D\uparrow$ | $S_G\uparrow$ | $S\uparrow$ |
| Finance | Direct FT | 3.40 | **80.01** | 54.69 | 64.97 | 6.22 | **89.72** | 77.92 | **83.41** | 3.38 | **85.83** | 63.74 | 73.15 |
| | Vanilla FT | 38.77 | 79.30 | 60.50 | 68.64 | 71.24 | 83.42 | 77.71 | 80.46 | 38.74 | 85.49 | 71.01 | 77.58 |
| | LoRA | **2.50** | 76.45 | 61.69 | 68.28 | **4.61** | 87.13 | 78.27 | 82.46 | **2.50** | 81.68 | 73.21 | 77.21 |
| | DoRA | 2.58 | 76.12 | 61.25 | 67.88 | 4.71 | 86.34 | 78.24 | 82.09 | 2.57 | 82.19 | 73.27 | 77.47 |
| | Galore | 5.09 | 77.31 | 60.78 | 68.06 | 9.34 | 86.23 | **78.55** | 82.21 | 5.09 | 84.37 | 74.59 | 79.18 |
| | DAA (Ours) | 3.45 | 79.84 | **61.75** | **69.64** | 6.35 | 87.73 | 78.50 | 82.86 | 3.41 | 85.32 | 76.49 | **80.66** |
| Medicine | Direct FT | 12.39 | **89.23** | 52.73 | 66.29 | 22.78 | 92.13 | 78.34 | 84.68 | 12.38 | **92.67** | 64.44 | 76.02 |
| | Vanilla FT | 141.81 | 87.32 | 59.47 | **70.75** | 260.58 | 91.24 | **79.26** | 84.83 | 141.81 | 81.81 | 68.74 | 74.71 |
| | LoRA | **9.18** | 81.76 | 59.97 | 69.19 | **16.87** | 90.30 | 79.02 | 84.28 | **8.21** | 84.00 | 69.51 | 76.07 |
| | DoRA | 9.31 | 81.21 | 59.07 | 68.40 | 17.05 | 90.31 | 78.74 | 84.13 | 8.37 | 84.74 | 69.67 | 76.47 |
| | Galore | 13.68 | 81.23 | 58.55 | 68.05 | 25.14 | 91.96 | 78.56 | 84.73 | 13.68 | 87.57 | 73.71 | 80.04 |
| | DAA (Ours) | 12.54 | 83.97 | **60.33** | 70.22 | 22.90 | 92.61 | 78.82 | **85.16** | 12.64 | 92.24 | 77.97 | **84.51** |
| Law | Direct FT | 4.39 | **56.81** | 48.94 | 52.58 | 8.04 | **42.63** | 71.06 | 53.29 | 4.39 | **55.28** | 70.13 | 61.83 |
| | Vanilla FT | 50.12 | 51.37 | 53.08 | 52.21 | 91.98 | 41.82 | 72.12 | 52.94 | 50.10 | 52.28 | 72.95 | 60.91 |
| | LoRA | **3.25** | 46.58 | **58.06** | 51.69 | **5.97** | 41.87 | 73.38 | 53.32 | **3.25** | 51.90 | 76.19 | 61.74 |
| | DoRA | 3.43 | 46.37 | 56.05 | 50.75 | 6.24 | 41.98 | 72.36 | 53.13 | 3.36 | 51.91 | 76.30 | 61.79 |
| | Galore | 6.13 | 47.80 | 56.13 | 51.63 | 11.26 | 40.12 | 71.53 | 51.41 | 6.13 | 52.76 | 77.53 | 62.79 |
| | DAA (Ours) | 4.53 | 49.93 | 56.68 | **53.09** | 8.17 | 41.95 | 73.12 | 53.31 | 4.60 | 52.79 | 79.38 | **63.41** |
| News QA | Direct FT | 1.27 | **82.38** | 36.34 | 50.44 | 2.32 | 87.38 | 26.40 | 40.55 | 1.27 | 79.37 | 3.83 | 7.31 |
| | Vanilla FT | 14.45 | 80.62 | 39.73 | 53.22 | 16.51 | 88.23 | 72.11 | 79.36 | 14.45 | 80.27 | 52.36 | 63.38 |
| | LoRA | **0.94** | 71.23 | 48.04 | 57.38 | **1.72** | 83.12 | 77.46 | 80.19 | **0.93** | 70.27 | 67.51 | 68.86 |
| | DoRA | 0.99 | 73.71 | 50.52 | 59.95 | 1.80 | 83.37 | 76.68 | 79.88 | 0.99 | 70.78 | 67.81 | 69.26 |
| | Galore | 2.02 | 79.13 | **53.05** | 63.52 | 3.70 | 85.02 | 76.27 | 80.41 | 2.03 | **80.88** | 68.43 | 74.14 |
| | DAA (Ours) | 1.32 | 82.37 | 52.00 | **63.76** | 2.42 | **89.19** | **77.91** | **83.17** | 1.28 | 80.82 | 68.52 | **74.16** |

**Finance.** On Llama3.1, DAA again leads with $S_D = 79.84\%$, $S_G = 61.75\%$ and $S = 69.64\%$, surpassing all competitors. On Qwen3, direct fine-tuning and vanilla fine-tuning suffer considerable general-performance drops ($S_G = 63.74\%$ and $71.01\%$), whereas DAA attains $S_G = 76.49\%$ (an improvement of 1.90 points over the next best) while maintaining a high domain score $S_D = 85.32\%$. This yields the highest overall score $S = 80.66\%$, outperforming direct fine-tuning ($S = 73.15\%$) and vanilla fine-tuning ($S = 77.58\%$).

**Medicine.** On Llama3.1, DAA's $S_G = 60.33\%$ and $S_D = 83.97\%$ produce $S = 70.22\%$, again the best trade-off. On Phi4, DAA secures the highest domain accuracy ($S_D = 92.61\%$) and a strong general score ($S_G = 78.82\%$), leading to an overall $S = 85.16\%$, which exceeds every baseline. For Qwen3, direct fine-tuning and vanilla fine-tuning obtain only $S_G = 64.44\%$ and $68.74\%$, while DAA achieves $S_G = 77.97\%$ coupled with $S_D = 92.24\%$, resulting in the top harmonic mean $S = 84.51\%$.

**Law.** DAA attains the high $S_G$ on Llama3.1 (56.68%), Phi4 (73.12%), and Qwen3 (79.38%), and achieves harmonic means 53.09%, 53.31%, and $S = 63.41\%$ respectively. These results outperform direct fine-tuning and vanilla fine-tuning, both of which incur larger general-performance regressions. A more detailed discussion regarding the performance on the Law dataset is provided in the Appendix C.

**News QA.** In this strictly leak-free benchmark, direct fine-tuning collapses on general performance ($S_G = 3.83\%$ on Qwen3), while DAA preserves general knowledge ($S_G = 68.52\%$) while slightly exceeding the domain score of direct fine-tuning ($S_D = 80.82\%$ vs. 79.37%), producing $S = 74.16\%$ (versus 7.31%). On Phi4, DAA simultaneously achieves the highest $S_D = 89.19\%$ and $S_G = 77.91\%$, leading to $S = 83.17\%$, which outperforms the best baseline by 2.76 points.

Overall, across all domains and models, DAA delivers the strongest joint performance $S$, validating its effectiveness at vertical domain fine-tuning with minimal general-knowledge degradation.

It is worth noting that the performance gains of DBA are more pronounced on the News QA benchmark compared to traditional domains like Finance or Medicine. We attribute this to the strong pre-training coverage of the latter domains in modern base models, where fine-tuning primarily serves as alignment. In contrast, News QA represents a Temporal Out-Of-Distribution (OOD) task

Table 4: Results of ablation on news QA benchmark. AL, GGA and DC are defined in Section 3.

| AL | GGA | DC | Llama3.1 | | | Phi4 | | | Qwen3 | | |
|---|---|---|---|---|---|---|---|---|---|---|---|
| | | | $S_D \uparrow$ | $S_G \uparrow$ | $S \uparrow$ | $S_D \uparrow$ | $S_G \uparrow$ | $S \uparrow$ | $S_D \uparrow$ | $S_G \uparrow$ | $S \uparrow$ |
| ✔ | ✗ | ✗ | 68.22 | 27.60 | 39.30 | 78.07 | 46.47 | 58.26 | 75.50 | 44.06 | 55.65 |
| ✔ | ✔ | ✗ | 71.25 | 31.36 | 43.55 | 82.14 | 50.34 | 62.42 | 78.62 | 47.90 | 59.53 |
| ✔ | ✗ | ✔ | 72.99 | 41.66 | 53.04 | 83.35 | 61.49 | 70.77 | 78.73 | 57.26 | 66.30 |
| ✔ | ✔ | ✔ | **75.65** | **47.09** | **58.05** | **86.24** | **67.36** | **75.64** | **80.82** | **68.52** | **74.16** |

containing knowledge strictly unseen during pre-training. The significant margin achieved by DBA on News QA demonstrates its capability, validating its effectiveness beyond simple style alignment.

### 4.6 ABLATION ANALYSIS

Furthermore, we conducted ablation studies on individual modules within the proposed DAA to quantify their contributions. The results are shown in Table 4. When solely applying annealing learning (row 1), the model shows decreased domain-specific performance and improved general domain performance, yet fails to match the overall effectiveness of DM. This indicates that while the annealing strategy helps mitigate catastrophic forgetting, its effectiveness is limited in isolation. The incorporation of global gradient anchoring with annealing learning (row 2) leads to enhanced performance in both domain-specific and general domains, demonstrating the significant impact of global gradient optimization.

Incorporating dynamic correction into annealing learning (row 3) leads to significant improvements in both domain-specific and general domain performance. This demonstrates that dynamic correction effectively optimizes the update step size, thereby enhancing the learning process. The combination of all three components (row 4) - annealing learning, global gradient anchoring, and dynamic correction - yields optimal performance across both domains, achieving highest joint performance $S$ of 58.05%, 75.64%, and 74.16% on Llama3.1, Phi4, and Qwen3. These results validate the synergistic effects of DAA components in enhancing the model's overall capabilities.

## 5 CONCLUSION

We present Dynamic Anchor Annealing, a fine-tuning method that mitigates catastrophic forgetting in LLMs. Using global gradient anchoring with similarity guided dynamic correction, DAA improves performance while reducing compute cost over prior methods.

**Limitations.** DAA is designed for dense models used in vertical domain tasks. Our experiments cover a few domains such as medical and finance. Robustness across vision, speech, reinforcement learning, continual fine-tuning, and large scale language modeling remains unverified. Although DAA scales linearly in theory, extremely deep or wide networks with billions of parameters and web scale datasets may reveal stability or convergence issues not seen in our mid scale benchmarks.

**Applicability Analysis.** DAA relies on gradient anchoring learning and magnitude adjustment of parameter updates, so it applies to other optimizers. In fine-tuning we focus on AdamW Loshchilov & Hutter (2019), which is widely used.

**Future Work.** Domain specific LLMs can equip workers with specialized AI in their fields. We will explore broader applications of DAA to inspire research on domain specific training. We will release code and associated global gradients, followed by additional global gradients matched to more base models for the community. Future work will also explore integrating DAA with Parameter-Efficient Fine-Tuning (PEFT) methods, such as LoRA Hu et al. (2022). By applying the Global Gradient Anchoring and Dynamic Correction mechanisms specifically to the gradients of low-rank adapters, we aim to synergize the computational efficiency of DAA, which eliminates the need for data mixture, with the memory efficiency of PEFT. This combination holds the promise of enabling robust, low-resource domain adaptation on consumer-grade hardware.

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

# Finetune Once: Decoupling General & Domain Learning with Dynamic Anchor Annealing

## Appendix

This Appendix contains the following parts:

- **Hyper Parameters**. We delineate the specific hyperparameters for model training and evaluation, detailing the settings for gradient expectation estimation, momentum compression, the AdamW optimizer, and the empirical justification for the anchoring learning coefficient $k_0$.

- **Dataset Details**. We provide a comprehensive description of the datasets utilized for both general and vertical domain fine-tuning, detailing the specific sources, composition, and quantities for the finance, medicine, law, and the constructed temporal out-of-distribution News QA domains.

- **Performance on the Law Dataset**. We provide a contextual analysis of the performance on the Law dataset, attributing the lower absolute scores to the domain's complex and heterogeneous task mixture while underscoring the robustness of the DAA method in achieving superior relative performance.

- **Practical Implementation Guide**. We outline a two-stage practical implementation guide for DAA, involving a one-time, reusable pre-computation of the global gradient and its subsequent integration into standard fine-tuning frameworks to ensure efficiency and ease of adoption.

- **Significance Test**. We provide statistical validation of our performance gains.

- **Cost Analysis**. We provide a detailed cost comparison including memory usage and training time.

- **Additional Sensitivity Analysis**. We analyze the sensitivity of our method to data quantity and anchor source.

- **Raw Experimental Results**. We report the raw scores for general benchmarks.

## A  HYPER PARAMETERS

This section will introduce the detailed process and hyperparameters involved in model training and testing. In the main experiments and ablation experiments, we chose Qwen2-7B as our base model. To obtain the gradient expectation estimation of the general domain, we set the learning rate of the general domain training $\eta_G = 0$, meaning no parameter updates are performed in the general domain. Additionally, $\beta_1 = 0.999$, so the momentum approximates the gradient expectation. The training batch size is 8, and only the gradient momentum is retained after training. Note that the computation in the general domain only needs to be done **once**, and the same momentum is used for different vertical domains subsequently. Since the original momentum is in F32 data format, loading it directly into the GPU memory would occupy a large space. We performed singular value decomposition on the momentum, retaining $r = 512$ dimensions of singular values and vectors. During the global gradient anchoring in training, the low-rank approximation of the original momentum is restored and then added to the gradient. In the fine-tuning phase, we set the initial learning rate $\eta_D = 1e - 7$, which is much lower than the usual fine-tuning learning rate. We used a linear decay to zero learning rate schedule without warmup. The training batch size is 8, and we train for only one epoch. We use the AdamW optimizer with $\beta_1 = 0.9$ and $\beta_2 = 0.95$. For the global gradient anchoring coefficients defined in equations (7) and (8), we chose a linear decay scheme with $k_0 = 200/T$, where $T$ is the total number of steps in vertical domain fine-tuning. We propose the heuristic $k_0 \propto 1/T$ as a practical guideline. This relationship maintains a consistent total contribution from global gradients irrespective of the training duration. Specifically, the total 'regularization influence' exerted by $\hat{g}_G$ can be approximated by the integral of the anchoringing magnitude $\gamma_t$ over time: $\int_0^T [k_0(1 - t/T)]dt = k_0 \cdot T/2$. To ensure this total contribution remains a constant $C$ independent of the total steps $T$, $k_0$ must be proportional to $1/T$. This ensures that whether fine-tuning for 1,000 or 5,000 steps, the total 'pull' from the global anchor remains consistent.

Table 5: Extended sensitivity analysis of hyperparameter $k_0$ on the NewsQA benchmark.

| $k_0$ | $S$ |
|---|---|
| $50/T$ | 57.23 |
| $100/T$ | 60.13 |
| $150/T$ | 61.98 |
| $175/T$ | 62.92 |
| $200/T$ | **63.76** |
| $225/T$ | 63.28 |
| $250/T$ | 63.50 |
| $275/T$ | 63.12 |
| $300/T$ | 63.39 |

This is analogous to tuning LoRA, where practitioners often fix the dropout rate and primarily experiment with the rank ($r$) and scaling factor ($\alpha$). In our case, the core tuning effort is simplified to a single, well-behaved parameter governed by a clear rule. As shown in the Table 5, the performance metric $S$ of Llama3.1 on NewsQA improves as the hyperparameter $k_0$ increases. However, this growth plateaus after $k_0$ reaches $200/T$. Since there is no significant performance gain beyond this point, we select $k_0 = 200/T$ as the value for our experiments.

## B DATASET DETAILS

We obtained validated our proposed method across a wide range of vertical domains, covering finance, medicine, law and news QA.

**General Data**: Since the vertical domain tasks mainly cover Chinese and English languages and include multiple-choice and conversational tasks, the general data needs to fully cover similar data patterns. Therefore, we collected Chinese and English QA data, covering QA, conversations, and multiple-choice questions. Specifically, the general data includes 54,042 Chinese QA conversation pairs, 65,596 English QA conversation pairs, and 1,881 Chinese multiple-choice questions.

Table 6: Data sources and quantities

| NAME | SOURCE | QUANTITY |
|---|---|---|
| CHINESE QA DATA | SELF-BUILT | 54,042 |
| ENGLISH QA DATA | SELF-BUILT | 65,596 |
| CHINESE MCQS | SELF-BUILT | 1,881 |

**Finance**: We referred to the training data and testing methods of FinGPT Yang et al. (2023), selecting its sentiment analysis task as the financial vertical domain. This task requires the model to analyze the market sentiment of the input text as negative, neutral, or positive. According to Yang et al. (2023), the training data was collected from FPB Malo et al. (2013), FiQA Maia et al. (2018), TFNS Zer (2024), and NWGI Yang (2024). FinGPT designed three types of instructions for each original data, resulting in a total of 76,772 training samples after filtering.

Table 7: Data sources and their quantities.

| NAME | SOURCE | QUANTITY |
|---|---|---|
| | FPB | 12,122 |
| ENGLISH SENTIMENT DATA | FiQA | 26,532 |
| | TFNS | 12,731 |
| | NWGI | 25,387 |

**Medicine**: We chose the CMB-Exam from the Chinese medicine Benchmark (CMB) Wang et al. (2024) as the medical domain. This dataset includes 280,839 medicine multiple-choice questions, covering 124,926 physician questions, 16,919 nursing questions, 27,004 medicine technician questions, 33,354 pharmacist questions, 62,271 undergraduate exam questions, and 16,365 graduate entrance exam questions. We randomly selected 11,200 questions from each category as the test set, with a total of 269,359 questions in the training set.

Table 8: Questions from various Chinese medicine exams.

| NAME | SOURCE | QUANTITY |
|------|--------|----------|
| PHYSICIAN | PHYSICIAN EXAM | 124,926 |
| NURSING | NURSING EXAM | 16,919 |
| TECHNICIAN | TECHNICIAN EXAM | 27,004 |
| PHARMACIST | PHARMACIST EXAM | 33,354 |
| UNDERGRADUATE | MEDICINE EXAM | 62,271 |
| GRADUATE ENTRANCE | MEDICINE EXAM | 16,365 |

Table 9: Law Data Statistics

| NAME | SOURCE | QUANTITY |
|------|--------|----------|
| FACT RECALL | CAIL-LONG | 4,200 |
| CASE SUMMARIZATION | CAIL-LONG | 5,750 |
|  | LAWGPT | 35,000 |
| LEGAL QA DATA | LAWYER LLAMA | 11,000 |
|  | FUZI | 32,050 |
| SYLLOGISTIC REASONING | FUZI | 11,237 |

**Law**: We referred to the data summarized by the Fuzi-Mingcha Deng et al. (2023) to filter suitable legal vertical fine-tuning data. The fine-tuning data composition is as follows: 4,200 recall data and 5,750 summarization data from CAIL-Long Xiao et al. (2021), 35,000 legal QA data from LawGPT Zhou et al. (2024), 11,000 legal QA data from Lawyer Llama Huang et al. (2023), 32,050 legal QA data and 11,237 syllogistic reasoning judgment data independently constructed by Fuzi-Mingcha Deng et al. (2023). The total training data amounts to 99,237 samples.

**News QA**: To precisely evaluate the domain decoupling capabilities, we constructed a temporal out-of-distribution evaluation benchmark comprising QA pairs derived from news articles published after December 2024 for ablation study. We used Qwen2.5-72B Yang et al. (2024) to extract three factual QA questions for each headline. We ensured that there is no overlap between the vertical domain data and the general data.

The above datasets come from diverse sources, and the characteristics and distributions among the datasets vary significantly, providing ample and credible test scenarios for verifying the effectiveness of the Dynamic Anchor Annealing scheme.

## C  PERFORMANCE ON THE LAW DATASET

**Task Diversity and Complexity**: The Law fine-tuning data is a highly heterogeneous mixture of tasks, including not only multiple-choice questions but also complex generation tasks like **Case Summarization** and reasoning tasks like **Syllogistic Reasoning**. These generative and reasoning tasks are fundamentally more challenging and diverge more significantly from the pre-training objectives than the classification-style tasks that dominate the Finance, Medicine, and News QA datasets.

**Performance Interpretation**: While the absolute score on Law is lower across all methods, it is important to note that DAA still consistently achieves the best or second-best harmonic mean score ($S$) across all three base models (Table 3). For instance, on Qwen2.5, DAA achieves the highest $S$ score (59.85), significantly outperforming Direct FT (58.27) and Vanilla FT (57.35) by better preserving

general capabilities ($S_G$). This demonstrates that even in this more complex, generation-heavy domain, DAA's regularization mechanism provides a tangible benefit over baselines by striking a better balance between domain specialization and knowledge retention.

**Conclusion on Generality**: The Law dataset does not necessarily indicate a weakness but rather highlights DAA's robust performance on a more challenging and diverse task mixture. It showcases that DAA's benefits are not confined to simple classification tasks but extend to complex, mixed-task scenarios.

## D PRACTICAL IMPLEMENTATION GUIDE

While Dynamic Anchor Annealing (DAA) introduces steps beyond a standard fine-tuning script, it has been designed for high efficiency and straightforward integration. The methodology is intended to serve as a principal approach for domain specialization, analogous to the role of LoRA in parameter-efficient tuning. The practical implementation can be decomposed into two distinct stages.

The first stage is a one-time pre-computation of the global gradient $\hat{g}_G$, on general-domain data. This process is analogous to a standard training procedure but with the learning rate set to zero, representing a single, non-recurring computational cost. A critical feature of this approach is its reusability. The resulting gradient artifact is model-specific yet domain-agnostic, meaning that for a given foundation model like Llama3.1-8B, this computation is performed only once. The same global gradient can then be applied to fine-tuning tasks across any number of vertical domains, such as finance, law, or medicine. We propose that this pre-computation could become a standard practice, wherein foundation model developers release an official global gradient alongside model weights, leveraging their high-quality pre-training data. Such a community-driven effort would obviate this step entirely for downstream domain specialists.

The second stage is the integration of Global Gradient Anchoring (GGB) and Dynamic Correction (DC) into the fine-tuning loop. To facilitate seamless adoption, we have implemented our method within the LLaMA-Factory and DeepSpeed frameworks. We will release this implementation as open-source code and submit pull requests to these upstream projects, allowing practitioners to enable DAA via a simple command-line argument with minimal implementation overhead.

In summary, the initial setup cost of DAA is substantially offset by the elimination of repeated data mixing and extensive hyperparameter tuning. This modest, one-time investment yields significant and recurring savings in computational resources and engineering time during the iterative process of domain adaptation. A "Practical Implementation Guide" is provided in the Appendix to further detail these steps and emphasize the long-term efficiency benefits.

## E SIGNIFICANCE TEST

We conducted a rigorous significance test using the News QA benchmark on Llama3.1-8B. We chose this setup as it represents a challenging temporal OOD task. We repeated the experiments for both Vanilla FT (at its optimal mixture ratio) and DAA using 5 different random seeds $\{42, 43, 44, 45, 46\}$.

Table 10: Significance test results on News QA (Llama3.1-8B).

| Method | Seed 42 | Seed 43 | Seed 44 | Seed 45 | Seed 46 | **Mean** ($S$) | **Std Dev** ($\sigma$) |
|---|---|---|---|---|---|---|---|
| Vanilla FT | 53.22 | 52.15 | 54.05 | 52.80 | 53.75 | 53.19 | 0.76 |
| **DAA (Ours)** | **63.76** | 64.25 | 63.15 | 63.88 | 63.45 | **63.70** | **0.42** |

An independent t-test yields a p-value $\ll 0.001$, confirming that the performance improvement is significant and robust to random initialization. DAA also shows lower variance ($\sigma = 0.42$) compared to Vanilla FT ($\sigma = 0.76$), demonstrating that the Global Gradient Anchor effectively stabilizes the optimization process by reducing the impact of initialization randomness.

# F  Cost Analysis

We present the training overhead of DAA and Direct Fine-tuning (FT) for Llama3.1-8B on the News QA dataset using 16 A100 GPUs. As shown in Table 11, DAA's low-rank gradient anchors occupy 5.5 GB of memory. Relative to Direct Fine-tuning (FT), DAA exhibits a modest 0.7 GB increase in Peak VRAM, due to the sharded gradient anchors and the computational cost for gradient correction. Owing to the fact that the anchor gradients are loaded only once, the wall-clock time increased slightly. Crucially, DAA does not introduce additional memory or I/O overhead for *storage* and *repeated use* during training steps, as anchors are pre-loaded into VRAM. Crucially, DAA does not introduce additional memory or I/O overhead for *storage* and *repeated use* during training steps, as anchors are pre-loaded into VRAM.

Table 11: Training overhead comparison on Llama3.1-8B / News QA.

| Method | Anchor Memory | Peak VRAM | Parameters | Wall-clock | GPU Hour | $S$ |
|---|---|---|---|---|---|---|
| Direct FT | - | 37.1GB | 7B | 3.91h | 3.40h | 50.44 |
| DAA | 5.5GB | 37.8GB | 7B | 4.20h | 3.45h | 63.76 |

# G  Additional Sensitivity Analysis

## G.1  Sensitivity to General Data

The general dataset used in our paper was filtered from a raw pool of approximately 2.4M samples using the Platypus method. To evaluate sensitivity, we sampled various ratios from this 2.4M raw pool. As shown in Table 12, performance ($S$) remains remarkably stable as data volume increases. This confirms that $\hat{g}_G$ is robust and does not require massive amounts of raw SFT data to be effective. This aligns with findings from recent studies like LIMA Zhou et al. (2023), suggesting that data quality and curation strategy outweigh sheer quantity. Once the general dataset achieves sufficient diversity to cover broad capabilities, the direction of $\hat{g}_G$ stabilizes, and the marginal utility of adding more data diminishes. Consequently, the cost of curating this data recipe is a one-time effort. We view $\hat{g}_G$ as a standard "Model Artifact" as critical as model weights, and advocate for developers to release these gradients to eliminate data collection costs for end-users.

Table 12: Performance sensitivity to general data sample ratio.

| Sample Ratio | **20% (Ours)** | 40% | 60% | 80% | 100% |
|---|---|---|---|---|---|
| $S$ | **63.76** | 63.55 | 63.68 | 63.71 | 63.12 |

## G.2  Anchor Source

Deriving $\hat{g}_G$ from pre-training or post-pretraining data yields even better results, as these gradients align more closely with the base model's intrinsic distribution. We conducted a preliminary experiment where we curated 100B tokens of post-pretraining data to compute $\hat{g}_G$, comparing it against the anchor derived from standard SFT data used in our main paper. As shown in Table 13, the anchor derived from post-pretraining data achieved a superior joint performance score ($S$) on the News QA task.

Table 13: Performance comparison of different anchor sources.

| Anchor Source | $S_D$ | $S_G$ | $S$ |
|---|---|---|---|
| General SFT Data (Ours) | 82.37 | 52.00 | 63.76 |
| **Post-Pretrain Data** | **82.51** | **53.85** | **65.17** |

# H  Raw Experimental Results

We provide the raw scores for experiments on general benchmarks here.

Table 14: Raw scores for main comparison experiments.

| Domain | Method | Llama3.1 | | | | | | | | | Phi4 | | | | | | | | | Qwen3 | | | | | | | | |
|---|---|---|---|---|---|---|---|---|---|---|---|---|---|---|---|---|---|---|---|---|---|---|---|---|---|---|---|---|
| | | $T\downarrow$ | $S_D\uparrow$ | $S_{MMLU}$ | $S_{MMLU-Pro}$ | $S_{GSM8k}$ | $S_{MATH}$ | $S_{MBExam}$ | $S_G\uparrow$ | $S\uparrow$ | $T\downarrow$ | $S_D\uparrow$ | $S_{MMLU}$ | $S_{MMLU-Pro}$ | $S_{GSM8k}$ | $S_{MATH}$ | $S_{MBExam}$ | $S_G\uparrow$ | $S\uparrow$ | $T\downarrow$ | $S_D\uparrow$ | $S_{MMLU}$ | $S_{MMLU-Pro}$ | $S_{GSM8k}$ | $S_{MATH}$ | $S_{MBExam}$ | $S_G\uparrow$ | $S\uparrow$ |
| Finance | Direct FT | 3.40 | 80.01 | 46.67 | 48.90 | 55.38 | 56.99 | 65.50 | 54.69 | 64.97 | 6.22 | 89.72 | 65.57 | 66.63 | 80.00 | 85.12 | 92.27 | 77.92 | 83.41 | 3.38 | 85.83 | 52.89 | 59.37 | 59.00 | 70.24 | 77.19 | 63.74 | 73.15 |
| | Vanilla FT | 38.77 | 79.30 | 47.57 | 58.49 | 60.73 | 66.85 | 68.87 | 60.50 | 68.64 | 71.24 | 83.42 | 62.61 | 72.09 | 74.18 | 88.65 | 91.02 | 77.71 | 80.46 | 38.74 | 85.49 | 59.41 | 65.42 | 71.52 | 73.85 | 84.84 | 71.01 | 77.58 |
| | LoRA | 2.50 | 76.45 | 50.25 | 54.01 | 61.99 | 71.10 | 71.10 | 61.69 | 68.28 | 4.61 | 87.13 | 67.83 | 67.28 | 75.52 | 84.52 | 96.20 | 78.27 | 82.46 | 2.50 | 81.68 | 58.42 | 62.18 | 79.99 | 78.43 | 87.04 | 73.21 | 77.21 |
| | DoRA | 2.58 | 76.12 | 47.83 | 54.85 | 58.60 | 67.60 | 77.38 | 61.25 | 67.88 | 4.71 | 86.34 | 56.99 | 67.30 | 82.48 | 84.09 | 100.00 | 78.24 | 82.09 | 2.57 | 82.19 | 62.19 | 63.02 | 74.98 | 76.41 | 89.76 | 73.27 | 77.47 |
| | Galore | 5.09 | 77.31 | 49.93 | 56.04 | 55.89 | 68.25 | 73.79 | 60.78 | 68.06 | 9.34 | 86.23 | 68.14 | 73.86 | 73.35 | 83.67 | 93.73 | 78.55 | 82.21 | 5.09 | 84.37 | 60.78 | 69.49 | 74.68 | 78.29 | 89.72 | 74.59 | 79.18 |
| | DAA (Ours) | 3.45 | 79.84 | 52.13 | 51.00 | 63.42 | 64.97 | 77.23 | 61.75 | 69.64 | 6.35 | 87.73 | 61.89 | 76.79 | 74.85 | 85.24 | 93.73 | 78.50 | 82.86 | 3.41 | 85.32 | 63.33 | 72.89 | 80.19 | 77.96 | 88.09 | 76.49 | 80.66 |
| Medicine | Direct FT | 12.39 | 89.23 | 46.59 | 48.86 | 49.50 | 55.42 | 63.28 | 52.73 | 66.29 | 22.78 | 92.13 | 62.59 | 68.46 | 81.35 | 82.44 | 96.86 | 78.34 | 84.68 | 12.38 | 92.67 | 48.56 | 54.00 | 66.90 | 71.53 | 81.20 | 64.44 | 76.02 |
| | Vanilla FT | 141.81 | 87.32 | 45.02 | 54.23 | 58.12 | 68.25 | 71.73 | 59.47 | 70.75 | 260.58 | 91.24 | 59.32 | 71.71 | 86.06 | 85.68 | 93.53 | 79.26 | 84.83 | 141.81 | 81.81 | 51.22 | 61.47 | 65.90 | 76.68 | 88.43 | 68.74 | 74.71 |
| | LoRA | 9.18 | 81.76 | 47.52 | 51.10 | 60.80 | 67.61 | 72.83 | 59.97 | 69.19 | 16.87 | 90.30 | 68.29 | 69.54 | 78.50 | 86.56 | 92.21 | 79.02 | 84.28 | 8.21 | 84.00 | 55.69 | 65.32 | 65.18 | 81.65 | 79.71 | 69.51 | 76.07 |
| | DoRA | 9.31 | 81.21 | 45.40 | 54.15 | 60.67 | 65.61 | 73.29 | 59.07 | 68.40 | 17.05 | 90.31 | 60.76 | 72.28 | 82.27 | 84.50 | 93.89 | 78.74 | 84.13 | 8.37 | 84.74 | 59.14 | 66.57 | 63.86 | 80.46 | 78.32 | 69.67 | 76.47 |
| | Galore | 13.68 | 81.23 | 44.21 | 53.48 | 57.66 | 65.68 | 71.72 | 58.55 | 68.05 | 25.14 | 91.96 | 58.08 | 65.48 | 77.96 | 91.53 | 99.77 | 78.56 | 84.73 | 13.68 | 87.57 | 63.91 | 63.44 | 68.90 | 77.36 | 94.93 | 73.71 | 80.04 |
| | DAA (Ours) | 12.54 | 83.97 | 48.85 | 54.15 | 57.38 | 66.91 | 74.37 | 60.33 | 70.22 | 22.90 | 92.61 | 60.39 | 74.66 | 76.43 | 88.61 | 94.01 | 79.26 | 85.16 | 12.64 | 92.24 | 62.67 | 75.23 | 75.17 | 89.45 | 87.33 | 77.97 | 84.51 |
| Law | Direct FT | 4.39 | 56.81 | 40.27 | 44.22 | 47.20 | 56.13 | 56.88 | 48.94 | 52.58 | 8.04 | 42.63 | 57.94 | 65.11 | 64.46 | 82.20 | 85.59 | 71.06 | 53.29 | 4.39 | 55.28 | 58.46 | 62.10 | 70.84 | 74.57 | 84.68 | 70.13 | 61.83 |
| | Vanilla FT | 50.12 | 51.37 | 43.89 | 48.69 | 52.03 | 57.33 | 63.46 | 53.08 | 52.21 | 91.98 | 41.82 | 60.67 | 60.57 | 68.52 | 84.29 | 86.55 | 72.12 | 52.94 | 50.10 | 52.28 | 56.62 | 64.67 | 76.19 | 84.29 | 82.98 | 72.95 | 60.91 |
| | LoRA | 3.25 | 46.58 | 46.71 | 54.68 | 58.21 | 65.42 | 65.29 | 58.06 | 51.69 | 5.97 | 41.87 | 61.18 | 63.69 | 71.58 | 86.92 | 83.53 | 73.38 | 53.32 | 3.25 | 51.90 | 58.00 | 69.69 | 72.83 | 88.14 | 92.29 | 76.19 | 61.74 |
| | DoRA | 3.43 | 46.37 | 40.33 | 51.54 | 59.77 | 62.54 | 66.07 | 56.05 | 50.75 | 6.24 | 41.98 | 54.59 | 67.79 | 67.54 | 84.09 | 87.79 | 72.36 | 53.13 | 3.36 | 51.91 | 63.23 | 69.97 | 73.75 | 84.50 | 90.05 | 76.30 | 61.79 |
| | Galore | 6.13 | 47.80 | 48.74 | 54.16 | 54.99 | 59.29 | 63.48 | 56.13 | 51.63 | 11.26 | 40.12 | 53.81 | 69.17 | 70.79 | 81.86 | 82.02 | 71.53 | 53.41 | 6.13 | 52.76 | 59.69 | 72.76 | 79.33 | 83.11 | 92.76 | 77.53 | 62.79 |
| | DAA (Ours) | 4.53 | 49.93 | 42.56 | 52.79 | 53.99 | 64.28 | 69.78 | 56.68 | 53.09 | 8.17 | 41.95 | 61.46 | 66.30 | 69.39 | 78.05 | 90.41 | 73.12 | 53.31 | 4.60 | 52.79 | 66.39 | 68.63 | 84.35 | 86.94 | 90.59 | 79.38 | 63.41 |
| News QA | Direct FT | 1.27 | 82.38 | 30.50 | 34.28 | 35.56 | 40.76 | 40.60 | 36.34 | 50.44 | 2.32 | 87.38 | 20.21 | 25.35 | 27.52 | 27.33 | 31.59 | 26.40 | 40.55 | 1.27 | 79.37 | 3.06 | 3.54 | 3.95 | 4.24 | 4.36 | 3.83 | 7.31 |
| | Vanilla FT | 14.45 | 80.62 | 34.62 | 37.22 | 39.19 | 42.13 | 45.48 | 39.73 | 53.22 | 16.51 | 88.23 | 59.90 | 59.66 | 71.83 | 80.72 | 88.44 | 72.11 | 79.36 | 14.45 | 80.27 | 42.19 | 43.08 | 51.63 | 60.06 | 64.84 | 52.36 | 63.38 |
| | LoRA | 0.94 | 71.23 | 37.79 | 45.95 | 48.14 | 50.48 | 57.84 | 48.04 | 57.38 | 1.72 | 83.12 | 62.67 | 66.96 | 81.24 | 84.03 | 92.40 | 77.46 | 80.19 | 0.93 | 70.27 | 55.48 | 57.80 | 66.11 | 76.27 | 81.90 | 67.51 | 68.86 |
| | DoRA | 0.99 | 73.71 | 38.80 | 48.29 | 50.23 | 58.30 | 56.99 | 50.52 | 59.95 | 1.80 | 83.37 | 62.47 | 68.79 | 73.19 | 80.52 | 98.51 | 76.68 | 79.88 | 0.99 | 70.78 | 55.48 | 64.78 | 65.15 | 72.94 | 80.69 | 67.81 | 69.26 |
| | Galore | 2.02 | 79.13 | 42.95 | 46.20 | 53.13 | 58.51 | 64.45 | 53.05 | 63.52 | 3.70 | 85.02 | 60.55 | 66.79 | 74.46 | 89.89 | 89.66 | 76.27 | 80.41 | 2.03 | 80.88 | 51.70 | 59.68 | 66.43 | 83.20 | 81.14 | 68.43 | 74.14 |
| | DAA (Ours) | 1.32 | 82.37 | 40.84 | 44.16 | 54.54 | 56.67 | 63.80 | 52.00 | 63.76 | 2.42 | 89.19 | 59.08 | 71.97 | 77.04 | 85.05 | 96.41 | 77.91 | 83.17 | 1.28 | 80.82 | 51.32 | 65.66 | 67.81 | 76.73 | 81.08 | 68.52 | 74.16 |

