# OpenReview forum: "Finetune Once: Decoupling General & Domain Learning with Dynamic Boosted Annealing"
_ICLR.cc/2026/Conference — ICLR 2026 Conference Withdrawn Submission_

### Official Review · Reviewer_T8Kj · 2025-10-26

**Soundness:** 2
**Presentation:** 3
**Contribution:** 2
**Rating:** 4
**Confidence:** 4

**Summary:**

This paper proposes a framework called Finetune-Once that aims to improve multi-domain fine-tuning efficiency of LLMs by decoupling the optimization of the model’s backbone and task-specific heads. The key idea is to first perform a shared backbone update that captures domain-general knowledge, followed by lightweight per-domain head updates to preserve domain-specific information. Experimennts on several benchmarks covering language understanding, reasoning, and coding tasks show that Finetune-Once achieves comparable or better performance than standard sequential or joint fine-tuning approaches, with reduced computational cost.

**Strengths:**

1. The problem of efficient multi-domain fine-tuning for LLMs is timely and practically relevant.
2. The proposed decoupling idea offers an intuitive way to separate shared and domain-specific optimization, potentially reducing interference.
3. The method is relatively simple to implement and could be useful in applied fine-tuning scenarios.

**Weaknesses:**

I have the following concerns. *If the authors could properly address them during the rebuttal phase, I am willing to raise my score.*

1. The idea of decoupling backbone and head optimization has been explored in several prior works, such as adapter-based learning, LoRA variants, and modular continual fine-tuning. The paper does not clearly distinguish how Finetune-Once differs conceptually or algorithmically from these existing methods.
2. This paper provides little analysis to justify why decoupling should prevent gradient interference or why the proposed optimization order (backbone first, head later) leads to better generalization.
3. Although the experiments show modest improvements, the gains are small and may fall within typical variance. No significance testing is provided.
4. Some important recent baselines [1,2,3] are missing. The paper would benefit from either a comparison with these methods or a discussion of their relevance.
5. The proposed Finetune-Once may underperform when domains are highly imbalanced or when tasks require cross-domain transfer, but the authors do not analyze such scenarios.

[1] Mixture-of-Skills: Learning to Optimize Data Usage for Fine-Tuning Large Language Models. EMNLP 2024.

[2] How Abilities in Large Language Models are Affected by Supervised Fine-tuning Data Composition. ACL 2024.

[3] Boosting Multi-Domain Fine-Tuning of Large Language Models through Evolving Interactions between Samples. ICML 2025.

**Questions:**

Please see Weaknesses.

---

> ### Author Response · Authors · 2025-11-20
>
> Thank you for your review. We appreciate you acknowledging that the problem of efficient fine-tuning is **"timely and practically relevant"** and that our decoupling idea offers an **"intuitive way"** to approach it.
>
> However, after carefully reading your summary, we believe there may be a **misunderstanding** regarding the core subject of our paper.
>
> W1-2. Misunderstanding of our paper
>
> We respectfully clarify a fundamental misunderstanding in the review. The reviewer characterizes our method as "decoupling the optimization of the model’s backbone and task-specific heads." We wish to emphasize that DBA does NOT employ backbone/head decoupling, nor does it freeze any model components.
>
> DBA is strictly a full-parameter fine-tuning framework where all weights are updated simultaneously. The "decoupling" in our title refers exclusively to the optimization level—specifically, decoupling the general learning signal from the domain-specific signal within the gradient space to prevent forgetting. It does not involve separating the model architecture or sequential training of backbones and heads.
>
> Nevertheless, we value your feedback and would like to address some of the general points you raised, as they are helpful for improving our paper.
>
> W3. On the significance of performance gains:
> We agree that in some scenarios, the margin of improvement for the S-Score may appear modest. However, we would like to emphasize the consistency of our method's performance. As shown in Table 3, DBA achieves the highest S-Score in 11 out of the 12 main experimental settings (spanning 4 domains and 3 base models). This demonstrates its robustness and reliability. To further validate these results, we will incorporate statistical significance tests in the final version to confirm that our observed gains are statistically significant.
>
> W4. On missing recent baselines:
> Thank you for pointing us to the three recent papers. We have reviewed them and found that their problem settings are different from ours. Specifically, [1] and [2] focus on optimizing the data mixture itself, while our work aims to eliminate the need for data mixing entirely. Paper [3] addresses a multi-domain fine-tuning scenario, whereas our work concentrates on the common single-domain adaptation challenge. While they are not direct baselines for our experiments, we agree they are important context. We will add a discussion of these papers to our related work section to better situate our contribution.
>
> W5. On the analysis of the method:
> Our "decoupling" mechanism works at the optimization level. By anchoring the domain-specific gradient with a pre-computed, stable global gradient, we regularize the update direction to prevent the model from deviating too far from its general capabilities. We will clarify this mechanism with a more intuitive explanation in the revised paper.
>
> We hope this clarifies the focus of our work. If you find the review was indeed for our paper, we are more than happy to discuss any of these points in greater detail.

---

> > ### Comment · Reviewer_T8Kj · 2025-11-21
> > **Thank you for the rebuttal.**
> >
> > Thank you for the rebuttal.
> >
> > The authors may want to update their submission so that a more complete and improved version of the paper can be evaluated. Once a revised paper is available, I will be able to assess more clearly whether the rebuttal adequately addresses my concerns. Without such updates, it is difficult to make an informed judgment based solely on a brief rebuttal.
> >
> > Best regards,
> >
> > Reviewer T8Kj

---

> ### Author Response · Authors · 2025-11-21
> **Responses to the Reviewer T8Kj [1/2]**
>
> Thank you for your prompt feedback. Due to the recommended rebuttal deadline, the validation experiments and manuscript revisions were still in progress at the time of submission. We now respond to your concerns further below.
>
> W3. On the significance of performance gains:
>
> We conducted a rigorous significance test using the News QA benchmark on Llama3.1-8B. We chose this setup as it represents a challenging temporal OOD task. We repeated the experiments for both Vanilla FT (at its optimal mixture ratio) and DBA using 5 different random seeds $\{42, 43, 44, 45, 46\}$.
>
> | Method | Seed 42 | Seed 43 | Seed 44 | Seed 45 | Seed 46 | **Mean ($S$)** | **Std Dev ($\sigma$)** |
> | :--- | :---: | :---: | :---: | :---: | :---: | :---: | :---: |
> | Vanilla FT | 53.22 | 52.15 | 54.05 | 52.80 | 53.75 | 53.19 | 0.76 |
> | **DBA (Ours)** | **63.76** | 64.25 | 63.15 | 63.88 | 63.45 | **63.70** | **0.42** |
>
> An independent t-test yields a p-value $\ll 0.001$, confirming that the performance improvement is significant and robust to random initialization. DBA also shows lower variance ($\sigma=0.42$) compared to Vanilla FT ($\sigma=0.76$), demonstrating that the Global Gradient Anchor effectively stabilizes the optimization process by reducing the impact of initialization randomness.
>
> The gains in domains like Finance, Medicine, and Law appear "modest" compared to News QA. We believe this phenomenon stems from the "saturation of base model capabilities". Modern foundational models (e.g., Llama3, Qwen) have likely encountered vast amounts of financial, medical, and legal data during pre-training. Consequently, fine-tuning in these areas acts more as "instruction alignment" or "recall" rather than learning entirely new knowledge, limiting the ceiling for improvement. This is precisely why we constructed the News QA benchmark. As detailed in Section 4.2, News QA consists of data strictly post-dating the base models' training cutoff (Temporal OOD). In this scenario, where the model must acquire new knowledge without catastrophic forgetting, DBA demonstrates a massive improvement (+10.54% in $S$ score on Llama3.1) compared to baselines. This confirms that DBA is exceptionally effective in scenarios requiring genuine domain adaptation.
>
> To clarify this point, we will add the following discussion to the Section 4.5:
>
> > "It is worth noting that the performance gains of DBA are more pronounced on the News QA benchmark compared to traditional domains like Finance or Medicine. We attribute this to the strong pre-training coverage of the latter domains in modern base models, where fine-tuning primarily serves as alignment. In contrast, News QA represents a Temporal Out-Of-Distribution (OOD) task containing knowledge strictly unseen during pre-training. The significant margin achieved by DBA on News QA demonstrates its capability, validating its effectiveness beyond simple style alignment."
>
> And, we would also like to clarify the core value proposition of DBA. Unlike methods solely focused on maximizing domain-specific metrics (often at the cost of general intelligence), our primary goal is to solve the "Efficiency-Performance-Generalization" trilemma in finetuning domain LLMs. To achieve domain performance comparable to or better than optimally tuned Vanilla FT, while (1) preserving general capabilities and (2) eliminating the massive computational cost of searching for data mixture ratios (reducing GPU hours by ~91%).

---

> ### Author Response · Authors · 2025-11-21
> **Responses to the Reviewer T8Kj [2/2]**
>
> W5: On the analysis of the method:
>
> Regarding your concern about "imbalanced domains" and "cross-domain transfer," we will add the following discussion to Section 2.1 of the revised manuscript:
>
> > "It is worth to differentiate the optimization challenges in DBA from those in Multi-Task Learning (MTL). Prior MTL works often focus on imbalanced domains, where high-resource tasks dominate the gradient direction at the expense of low-resource tasks [4], and cross-domain transfer, which aims to leverage knowledge from a source domain to improve a distinct target domain [5]. In contrast, DBA targets Finetuning to Target Domain , where the optimization landscape is defined by the tension between the General Prior (comprising a wide range of distributed domains) and the Target Domain (the fine-tuning task). (1) Handling Imbalance: Unlike MTL, where multiple specific domains compete for model capacity, the "imbalance" in our setting exists between the massive pre-training distribution and the smaller fine-tuning dataset. DBA explicitly regulates this imbalance through the Global Gradient Anchor and the decay coefficient $\gamma_t$, ensuring that the general prior is not overwhelmed by the specific domain, regardless of the domain's data scale. (2) Nature of Transfer: DBA focuses on General-to-Specific retention rather than Specific-to-Specific transfer. The Global Gradient serves as a regularization term that enforces the preservation of general reasoning structures while allowing the model to adapt to the target distribution. Thus, the negative transfer or interference often seen between disjoint domains in MTL is mitigated by anchoring optimization to the robust general manifold."
>
> *[4] Sener et al., Multi-Task Learning as Multi-Objective Optimization, NeurIPS 2018.*
>
> *[5] Liu, H., Long, M., Wang, J. & Jordan, M. Transferable Adversarial Training: A General Approach to Adapting Deep Classifiers. ICML 2018. *
>
>
> We hope our response and the revised paper could address your concerns. We are happy to engage in further discussion.
>
> Best regards,
>
> The Authors

---

> ### Comment · Reviewer_T8Kj · 2025-11-24
> **Thank you for the further response.**
>
> Thank you for the further response. After reading the rebuttal and the revised paper, most of my concerns have been addressed.
>
> However, **I am still *curious* about the performance of the previously mentioned works [1,2,3] under the setting of this paper**. For example, could the proposed method incorporate RL techniques for sample management, similar to [1]? Although [3] targets multi-domain scenarios, the method itself operates at the sample level rather than the domain level (and this is one of the key insights of [3]).
>
> I understand that adding experiments for other methods---especially those somewhat distant from the main focus of this paper---can be challenging during the rebuttal phase. Therefore, it would be sufficient for the authors to include some discussion in the paper. I believe that adding such discussion would greatly enhance the impact and quality of the work.
>
> In addition, the paper itself is interesting. Hence, I encourage other reviewers to join the discussion to help the authors further improve the paper.
>
> ---
>
> **References:**
>
> [1] Mixture-of-Skills: Learning to Optimize Data Usage for Fine-Tuning Large Language Models. EMNLP 2024.
>
> [2] How Abilities in Large Language Models are Affected by Supervised Fine-tuning Data Composition. ACL 2024.
>
> [3] Boosting Multi-Domain Fine-Tuning of Large Language Models through Evolving Interactions between Samples. ICML 2025.

---

> > ### Author Response · Authors · 2025-11-25
> > **Response to Follow-up Questions on RL Integration and Sample-Level Methods**
> >
> > We sincerely thank you for your continued engagement and positive assessment of our work. We appreciate the opportunity to discuss the relationship between our work and the recent studies [1, 2, 3]. We have added a detailed discussion to the Related Work section (Section 2.1) of the revised manuscript.
> >
> > Regarding RL for sample management [1]:
> >
> > DAA and methods like Mixture-of-Skills [1] operate at different levels of the optimization pipeline. While DAA focuses on optimization dynamics (gradient anchoring) to eliminate the need for searching data mixture ratios, it does not inherently judge data quality. Therefore, DAA is highly compatible with data selection techniques. Future work could indeed incorporate RL-based curricula to select the high-quality general data used to compute the Global Gradient Anchor ($\hat{g}_G$), potentially further enhancing the anchor's robustness. We have added this point to our manuscript.
> >
> > Regarding sample-level interactions [3]:
> >
> > We acknowledge that [3] offers a powerful mechanism for multi-domain coordination by modeling fine-grained, pairwise interactions between samples. This allows for dynamic adjustments based on how specific samples influence one another during training.  In contrast, DAA deliberately adopts a domain-level approach. Instead of modeling complex sample-to-sample dynamics, we abstract the entire general domain into a single, stable Global Gradient Anchor ($\hat{g}_G$). This design choice prioritizes computational efficiency and memory scalability. While [3] offers fine-grained control, calculating evolving interactions can incur significant overhead in large-scale scenarios. DAA’s domain-level anchor effectively regularizes the optimization trajectory with minimal cost, enabling a streamlined "Finetune Once" workflow that is accessible even under resource-constrained settings.
> >
> > We hope these clarifications and the added discussion in the manuscript fully address your concern. We deeply appreciate your constructive guidance and your initiative in encouraging further discussion among the reviewers. We remain fully open to any additional feedback that can help us further polish the quality and impact of this work.
> >
> > [1] Mixture-of-Skills: Learning to Optimize Data Usage for Fine-Tuning Large Language Models. EMNLP 2024.
> >
> > [3] Boosting Multi-Domain Fine-Tuning of Large Language Models through Evolving Interactions between Samples. ICML 2025.

---

> > > ### Comment · Reviewer_T8Kj · 2025-11-25
> > > **Thank you for the detailed follow-up response.**
> > >
> > > Thank you for the detailed follow-up response. I appreciate the authors' efforts in addressing my earlier questions and in clarifying the relevance of the cited works.
> > >
> > > After reviewing the latest version of the manuscript, I noticed that the discussion of [1] and [3]---which the authors indicated would be added---does not appear to have been incorporated into the paper. Including a concise discussion of these works would improve the completeness of the contextualization and help readers better understand how this paper situates itself within the broader literature.
> > >
> > > I encourage the authors to add these discussions in the next revision, as it would further strengthen the clarity and contribution of the paper.
> > >
> > > [1] Mixture-of-Skills: Learning to Optimize Data Usage for Fine-Tuning Large Language Models. EMNLP 2024.
> > >
> > > [2] How Abilities in Large Language Models are Affected by Supervised Fine-tuning Data Composition. ACL 2024.
> > >
> > > [3] Boosting Multi-Domain Fine-Tuning of Large Language Models through Evolving Interactions between Samples. ICML 2025.

---

> > > > ### Author Response · Authors · 2025-11-25
> > > > **Response to the Reminder on Missing Discussions**
> > > >
> > > > We sincerely thank you for your careful review and for bringing this to our attention. We apologize for the confusion; due to an unintentional oversight in the file upload process, the updated content was not correctly reflected in the previous version. We have immediately rectified this and ensured that the discussion regarding [1-3] is now fully incorporated into the manuscript. We deeply appreciate your patience and your keen eye, which ensures the completeness and quality of our work.
> > > >
> > > > [1] Mixture-of-Skills: Learning to Optimize Data Usage for Fine-Tuning Large Language Models. EMNLP 2024.
> > > >
> > > > [2] How Abilities in Large Language Models are Affected by Supervised Fine-tuning Data Composition. ACL 2024.
> > > >
> > > > [3] Boosting Multi-Domain Fine-Tuning of Large Language Models through Evolving Interactions between Samples. ICML 2025.

---

> > > > > ### Comment · Reviewer_T8Kj · 2025-11-25
> > > > > **Thank you for your continued and timely responses.**
> > > > >
> > > > > Thank you for your continued and timely responses. I have updated my score to 6. Addressing the remaining concerns raised by the other reviewers would further strengthen the paper.

---

> > > > > > ### Author Response · Authors · 2025-11-26
> > > > > > **Thank you for your support and constructive feedback**
> > > > > >
> > > > > > Dear Reviewer T8Kj,
> > > > > >
> > > > > > We are sincerely grateful for your decision to raise the score and for your engagement throughout the discussion period. Your constructive feedback—particularly regarding the distinction from MTL and the inclusion of relevant literature—has significantly improved the clarity and positioning of our work.
> > > > > >
> > > > > > We have taken note of your advice to address the remaining concerns from other reviewers and will continue to polish the manuscript to ensure it meets the highest standards. Thank you again for your time and valuable insights.
> > > > > >
> > > > > > Best regards,
> > > > > >
> > > > > > The Authors

---

### Official Review · Reviewer_aLnJ · 2025-10-27

**Soundness:** 3
**Presentation:** 4
**Contribution:** 3
**Rating:** 6
**Confidence:** 4

**Summary:**

This paper addresses a core, pervasive problem in large language model (LLM) fine-tuning: catastrophic forgetting of general capabilities when adapting to a specific domain, and the high computational cost of data mixture strategies used to mitigate it.
Experimental results show that DBA significantly outperforms baselines (including direct fine-tuning, vanilla fine-tuning with data mixture, and PEFT methods like LoRA) across four different domains (Finance, Medicine, Law, News QA) and three base models (Llama3.1, Phi4, Qwen3). DBA achieves the best balance between domain performance (SD) and general performance (SG), as measured by the harmonic mean (S). Most critically, because DBA completely eliminates the need for data mixture and its associated tuning, it is claimed to reduce GPU hours by up to 91.0% compared to the vanilla fine-tuning method.

**Strengths:**

The paper tackles a very practical and costly problem. In practice, finding the optimal data mixture ratio for each domain is a compute-intensive, empirical process. DBA offers a principled solution that claims to eliminate this process entirely, which is highly significant for the practical deployment and iteration of LLMs. The claimed 91.0% reduction in computational cost is a massive advantage.
The paper is exceptionally well-written. The problem statement (Table 1, Fig 1), methodology (Fig 3, Sec 3), and experimental results (Table 3) are all presented with great clarity.

**Weaknesses:**

The term "Gradient Boosted" is highly misleading. As the authors acknowledge in Section 3.1, this is not the standard meaning of sequentially fitting models to residuals, as in Gradient Boosting Machines (e.g., XGBoost). Here, it functions more as "gradient anchoring" or "gradient augmentation." Using a term with a well-established, different meaning in machine learning is likely to cause unnecessary confusion.

**Questions:**

DBA is presented as a full-parameter tuning technique. Have the authors investigated combining it with PEFT methods like LoRA? For example, applying the GGB and DC gradient modifications to the low-rank adapter instead of the full weights? This seems like a very promising direction to achieve both computational (no data-mixture) and memory (parameter-efficient) efficiency.

In Section 3.1, the derivation for GGB (Eq. 4) suggests that using a fixed $g_G$ reduces the variance of the combined gradient. This is intuitive. How does this connect to the motivation in Section 2.2 that the general domain (Fig 2a) has "many local optima"? Does GGB help the optimizer "bypass" these poor local optima by providing a smoother, more stable gradient (derived from $g_G$)? More intuition on how GGB addresses the observed loss landscape issue would be appreciated.

---

> ### Author Response · Authors · 2025-11-20
> **Responses to the Reviewer aLnJ [1/1]**
>
> Thank you for your positive and insightful review. We are delighted that you found our paper **"exceptionally well-written"** and highlighted that it **"tackles a very practical and costly problem"** with a **"massive advantage"** in cost reduction. We agree with the points you've raised and would like to respond to them here.
>
> W1. On the term "Gradient Boosted"
>
> We agree that the term “Gradient Boosted” is misleading due to its established meaning in the ML literature. Our initial intention was simply to indicate that the domain gradient is adjusted using information from the global gradient.
>
> To avoid confusion, we will rename the method to Global Gradient Anchoring (GGA) in the final version and update the text accordingly.
>
> Q1. On combining DBA with PEFT methods
>
> This is an excellent suggestion. Applying our method to PEFT is feasible: the anchoring and correction can be computed on the gradients of the adapter parameters rather than the full model weights.
>
> This would preserve the computational benefits of DBA while also reducing memory usage. We plan to include this as a direction for future work in the revised manuscript.
>
> Q2. On how GGB addresses the loss landscape issue
>
> Yes, there is a direct connection here, and GGB was designed precisely to address this issue.
>
> First, regarding the connection between the landscape (Fig 2a) and our method (Eq. 4) is: the rugged, multi-optima landscape observed in Fig 2a is a direct visual manifestation of high gradient variance.
>
>  The stochastic gradient $g_{G,t}$ computed on any given mini-batch from the general domain is a noisy estimate, and this noise is what creates the tortuous optimization path. Our GGB method (Eq. 4) tackles this root cause. By replacing the noisy, random variable $g_{G,t}$ with a fixed, pre-computed global gradient $\hat{g}_G$, we mathematically eliminate this source of variance from the general domain, thereby "smoothing" the overall optimization signal.
>
> Secondly, GGB dose help the optimizer 'bypass' these poor local optima by providing a smoother, more stable gradient. The global gradient $\hat{g}_G$ acts as a stable anchor. During fine-tuning, the domain gradient $g\_{D,t}$ pulls the model towards a domain-specific optimum, leading to catastrophic forgetting. The $\hat{g}_G$ component provides a constant, regularizing pull towards the "center of mass" of general knowledge. This corrective force prevents the optimizer from fully descending into valleys that are only good for the domain task.
>
> Finally, to offer more intuition beyond the mathematical and experimental formulations, consider a simple analogy. The finetuning process is like a ship trying to reach a specific destination. The domain gradient acts as the force pulling the ship forward. The high variance of the general domain, however, is like a turbulent, stormy sea, constantly knocking the ship off its course. Traditional data mixture methods are akin to constantly fighting the storm by making frequent course corrections. Our GGB method, in contrast, is like dropping a heavy anchor ($\hat{g}_G$). This anchor does not stop the forward pull, but it provides immense stability. It allows the ship to travel along a much smoother path toward its intended destination—a robust solution balancing both worlds—instead of being tossed by the storm to an unknown location (a poor local optimum).
>
> We hope this explanation addresses your questions. Thank you once again for your insightful feedback.

---

> > ### Author Response · Authors · 2025-11-26
> > **Gentle Reminder regarding our Rebuttal**
> >
> > Dear Reviewer aLnJ,
> >
> > Thank you again for your positive assessment and suggestions. As the discussion period is coming to a close, we wanted to gently follow up to ensure our response addressed your comments.
> >
> > Your suggestion to rename the method has improved the clarity of our presentation and eliminated potential terminological confusion. Per your advice, we have officially renamed the method from "Dynamic Boosted Annealing" to "Dynamic Anchor Annealing (DAA)" throughout the revised manuscript.
> >
> > Regarding your insightful question about PEFT, we have also clarified that DAA is compatible with low-rank adapters (applying anchoring to adapter gradients), which we have added as a promising future direction in the paper.
> >
> > We hope these revisions meet your expectations and would appreciate any further feedback you might have.
> >
> > Best regards,
> >
> > The Authors

---

### Official Review · Reviewer_PtTU · 2025-10-29

**Soundness:** 3
**Presentation:** 3
**Contribution:** 2
**Rating:** 4
**Confidence:** 4

**Summary:**

This paper introduces Dynamic Boosted Annealing (DBA), a novel fine-tuning framework designed to address catastrophic forgetting of general capabilities when adapting Large Language Models (LLMs) to specific domains. The authors aim to overcome the inefficiency of traditional data mixture strategies, which require costly repeated experiments to find optimal mixing ratios. DBA proposes a two-stage approach: first, it pre-computes a "global gradient" representing the general domain optimization direction using zero-learning-rate training; second, during fine-tuning on only domain-specific data, it uses this global gradient to "boost" the domain gradient (Global Gradient Boosted Learning - GGB), dynamically adjusts the update magnitude based on the similarity between global and domain gradients (Dynamic Correction - DC), and employs a very low, decaying learning rate (Annealing Learning - AL). The authors claim this "finetune once" approach significantly reduces computational cost while effectively balancing domain performance and general capability retention, presenting experiments across several models and domains.

**Strengths:**

1. The framework offers a highly practical and efficient alternative to data mixing for domain adaptation. By pre-calculating a reusable global gradient and enabling fine-tuning solely on domain data, DBA directly tackles the major cost bottleneck of repeated mixture experiments .



2. The proposed combination of techniques (GGB, DC, AL) represents an innovative attempt to explicitly decouple and balance general knowledge retention and domain-specific learning within the optimization process itself .





3. The empirical results demonstrate that DBA achieves a strong balance between domain performance and general capability preservation across multiple base models and domains, often outperforming baselines in harmonic mean score while drastically reducing GPU hours compared to vanilla fine-tuning .

**Weaknesses:**

1. The theoretical justification for Global Gradient Boosting (GGB) appears insufficient. While motivated by reducing variance, adding a fixed, pre-computed global gradient (potentially outdated as the model parameters change) to the domain gradient throughout training lacks rigorous analysis on why this acts as an effective regularizer, especially compared to established techniques. The connection drawn to SVRG seems tenuous given the fixed nature of $\hat{g}_G$.

2. The Dynamic Correction (DC) mechanism lacks clear intuition and adequate ablation. The paper adjusts update steps based on gradient similarity , potentially increasing step size when gradients differ  (based on Eq 9), but the rationale for why divergence warrants larger steps isn't fully explained. Furthermore, the ablation study (Table 4) only shows combined effects, not the isolated contribution or potential negative impacts of DC itself.

3. The performance comparison might overstate practical gains relative to optimally tuned baselines. While DBA significantly reduces the cost of finding the best data mixture, its final domain performance ($S_D$) may still be slightly below that achieved by vanilla fine-tuning at its optimal (but expensive to find) mixing ratio. The claimed 91% cost saving compares DBA to the entire process of searching for the best vanilla ratio, not a single run with the optimal ratio.

**Questions:**

1. Figure 2 is cited as Figure 3 in the text. Please correct the figure numbering or citation.

2. The calculation of the global gradient $\hat{g}_G$ uses $\beta_1 \to 1$. Does this require a full pass over the general data? How sensitive is $\hat{g}_G$ (and thus DBA's performance) to the size and quality of the general data used for this pre-computation?

3. The choice of $k_0 = 200/T$ for the GGB boost magnitude seems empirically derived. Is there any theoretical intuition for why this inverse relationship with total steps (T) is appropriate?

---

> ### Author Response · Authors · 2025-11-20
> **Responses to the Reviewer PtTU [1/2]**
>
> Thank you for your valuable feedback. We are encouraged that you recognized our framework as a **"highly practical and efficient alternative to data mixing"** and an **"innovative attempt"** to balance performance while **"drastically reducing GPU hours."**
>
> W1: Insufficient theoretical justification for GGB.
>
> We acknowledge that the analogy between GGB and SVRG might be imprecise, given that our global gradient $\hat{g}_G$ is fixed. We will revise this discussion in the updated manuscript and strengthen the motivation for GGB from the following two perspectives:
>
> 1.  Variance Reduction Intuition: While not a dynamic variance reduction mechanism, the core idea of GGB is to regularize each mini-batch computed, high-variance domain gradient ($g_{D,t}$) by introducing a fixed, low-variance global direction vector ($\hat{g}_G$ is obtained by averaging over a large amount of general data). As shown in Equation (4), the variance of the combined gradient originates solely from the domain gradient term and is scaled by $(1-\gamma_t)^2$. This leads to a more stable optimization process, preventing the severe oscillations that might occur when solely dominated by domain gradients, thereby preserving generalization capability.
>
> 2.  Smoothening the Optimization Trajectory: As illustrated in Figure 2(a), the loss surface of the general domain exhibits numerous local optima. $\hat{g}_G$ can be viewed as a 'compass' pointing towards a region of good generalization performance. When exploring in the 'dark forest' with only domain-specific data, this compass ensures that each update step is at least partially directed towards a favorable direction for generalization, thus circumventing local optima that are beneficial for domain-specific tasks but detrimental to general tasks.
>
> We will elaborate on these two points with clearer language and more direct argumentation, replacing the original SVRG analogy.
>
> W2: Dynamic Correction (DC) mechanism lacks clear intuition and adequate ablation
>
> This is a crucial point. The core design principle of DC is to dynamically adjust the learning step based on the interference strength that the fine-tuning task imposes on the model's general knowledge. This interference strength is precisely quantified by $s_t$, defined as the absolute cosine similarity between the gradients derived from the general domain and those from the fine-tuning task.
>
> When $s_t$ approaches zero, it indicates that the gradients are nearly orthogonal, implying a minimal impact of the fine-tuning task on the model's prior knowledge (e.g., in our News QA task). Under these conditions, the model can be updated with a larger learning rate.
>
> When $s_t$ approaches one, it indicates a high correlation between the fine-tuning task and the model's general knowledge (e.g., specific knowledge in medical or financial domains). In this scenario, the model requires only minor adjustments. An excessively large step size could potentially degrade the already acquired general knowledge.
>
> Furthermore, relying solely on DC is impractical. This is because direct FT without annealing learning often leads to a collapse in general performance, thereby obscuring the potential gains of DC. In our experiment, a comparison of rows 1 and 3 in the paper demonstrates that the introduction of DC achieved an average improvement of 12.3% in $S$.
>
> W3: Performance comparison might overstate practical gains
> Indeed, your observation is precise. The 91% cost savings reported in the paper are achieved by contrasting a single run of DBA with the complete Vanilla Fine-Tuning (FT) process, which necessitates multiple experiments to identify the optimal data mixing ratio. We contend that this represents DBA's paramount advantage: in real-world scenarios, the iterative search for an optimal data blending ratio for each new domain is an unavoidable and significant cost driver. By entirely obviating this procedure, DBA yields substantial efficiency improvements.

---

> ### Author Response · Authors · 2025-11-20
> **Responses to the Reviewer PtTU [2/2]**
>
> Q1: Error in Figure Citation
>
> We appreciate your keen observation. We have rectified the incorrect reference to Figure 3.
>
> Q2: On the Computation and Sensitivity of the Global Gradient $\hat{g}_G$
>
> Yes, computing $\hat{g}_G$ requires a full pass (one epoch) over the general dataset. Regarding the sensitivity of $\hat{g}_G$ to the general data (quantity and quality), we offer the following two perspectives:
>
> Diversity Outweighs Quantity: In the SFT setting, our findings regarding $\hat{g}_G$ acquisition align with conclusions from recent studies like LIMA [1]: diversity and quality are far more critical than sheer quantity. We observed that once the general dataset achieves sufficient diversity to cover broad capabilities, the direction of $\hat{g}_G$ stabilizes, and the marginal utility of adding more data diminishes. This implies that massive datasets are not strictly necessary to obtain a robust $\hat{g}_G$. Crucially, the cost of curating this data recipe is a one-time effort borne by the "gradient publisher." Most end-users can simply reuse this released gradient to enjoy the generalization safeguards without incurring the data preparation costs. We also provide a Sample Ratio Sensitivity table in our response to Reviewer UAVW (W5 & Q4), which demonstrates that performance remains stable even when training on partial subsets of data. This analysis will be added to the Appendix.
>
> Superiority of Pre-training/Post-pretraining Data: Furthermore, deriving $\hat{g}_G$ from pre-training or post-pretraining data yields even better results, as these gradients align more closely with the base model's intrinsic distribution. We conducted a preliminary experiment where we curated 100B tokens of post-pretraining data to compute $\hat{g}_G$, comparing it against the anchor derived from standard SFT data used in our main paper. As shown in the table below, the anchor derived from post-pretraining data achieved a superior joint performance score ($S$) on the News QA task. In light of this, we advocate for a community effort where model developers release official "General Gradient Anchors" alongside their base models.
>
> | Anchor Source | $S_D$ | $S_G$ | $S$ |
> | :--- | :---: | :---: | :---: |
> | General SFT Data (Ours) | 82.37 | 52.00 | 63.76 |
> | **Post-Pretrain Data** | **82.51** | **53.85** | **65.17** |
>
> *Note: Preliminary results on Llama3.1-8B / News QA.*
>
> [1] Zhou, C., et al. "Lima: Less is more for alignment." NeurIPS (2024).
>
> Q3: The theoretical intuition behind the choice of $k_0$ in GGB.
>
> The choice of $k_0$ indeed relies on a core theoretical intuition: the total 'regularization influence' exerted by the global gradient anchor $\hat{g}_G$ throughout the fine-tuning process should be a stable value independent of the total training steps $T$.
>
> The contribution of $\hat{g}_G$ at each step $t$ is determined by the coefficient $\gamma_t = k_0 * (1 - t/T)$. Thus, over a total of $T$ training steps, its cumulative contribution can be approximated by the integral of $\gamma_t$ from $t=0$ to $T$, i.e., $\int_0^T [k_0(1-t/T)]dt \approx k_0 \cdot T / 2$. For this total contribution $k_0 \cdot T / 2$ to be a constant $C$ independent of $T$, $k_0$ must be proportional to $1/T$, i.e., $k_0 \propto 1/T$.
>
> This simple inverse relationship ensures that, regardless of whether we fine-tune a domain for 1000 or 5000 steps, the total 'pull' received by the model from the global gradient anchor is approximately the same, thereby leading to better generalization and robustness for this hyperparameter. We will supplement the paper with this more detailed explanation.

---

> > ### Author Response · Authors · 2025-11-26
> > **Gentle Reminder regarding our Rebuttal**
> >
> > Dear Reviewer PtTU,
> >
> > We hope this message finds you well. We wanted to gently follow up to ensure you have had a chance to review our response and the revised paper before the discussion period ends.
> >
> > Based on your insightful feedback, we have strengthened the theoretical justification for Global Gradient Boosting (GGB) in the revision, moving away from the SVRG analogy to a clearer explanation based on variance reduction and optimization trajectory smoothing. We also provided further intuition and ablation details for the Dynamic Correction (DC) mechanism.
> >
> > Your feedback has been instrumental in refining our theoretical framework, and we hope the revised manuscript now aligns better with your standards. We remain fully available to answer any further questions you may have.
> >
> > Best regards,
> >
> > The Authors

---

### Official Review · Reviewer_UAVW · 2025-11-01

**Soundness:** 2
**Presentation:** 2
**Contribution:** 2
**Rating:** 4
**Confidence:** 4

**Summary:**

The paper proposes **Dynamic Boosted Annealing (DBA)**, a two-stage *full-parameter fine-tuning* framework intended to mitigate catastrophic forgetting without mixing in-domain and general data during training.

**Stage 1 (Zero-LR Global Gradient Estimation):** On a “general” dataset, the model is run with learning rate ≈ 0 and large β₁ so that Adam’s first-moment accumulator approximates the **expected general gradient**; this “anchor” gradient is then **SVD-compressed** (low rank) and stored.

**Stage 2 (Domain FT with Anchoring):** During domain fine-tuning, each step linearly combines the current domain gradient with the stored **global anchor gradient** (with a decaying weight), applies a **similarity-based dynamic correction**, and uses **very small initial LR + annealing**.

Evaluation uses a **balance metric** $S = \mathrm{HM}(S_D, S_G)$ (harmonic mean of domain and general scores). The paper reports higher $S$ than direct full FT and several PEFT baselines on multiple base models, plus experiments on a **temporal OOD** News-QA benchmark.

**Strengths:**

**Mitigation of Catastrophic Forgetting:** Compared to standard full fine-tuning (Direct FT), the core contribution of DBA is its ability to mitigate catastrophic forgetting. By **leveraging global gradient information** ($\hat{g}_{G}$) as a regularizing anchor during domain training , the method can specialize in a new domain while **maintaining a certain level of performance on general tasks** ($S_G$).

**Achieving a Performance Balance (S-Score):** The method is designed to optimize for a **balance** between domain-specific performance ($S_D$) and general capability retention ($S_G$). By preventing the collapse of $S_G$ that plagues Direct FT, DBA consistently achieves a superior harmonic mean ($S$) score, demonstrating its effectiveness in producing a more robust and practical model.

**Reusable global signal:** The **single** global anchor (after low-rank compression) can be reused across domains for the *same* base model, which is practically appealing for organizations that repeatedly fine-tune on related domains.

**Weaknesses:**

- **Cost fairness is not established.** The paper primarily compares **GPU-hours** (time) against **PEFT** baselines (LoRA/DoRA/etc.). That’s **not an apples-to-apples comparison** because **PEFT methods are designed mainly to be memory-efficient and to train with few tunable parameters**.
    - DBA is full fine-tuning **plus** an **extra memory/IO footprint** for storing and repeatedly using the low-rank **anchor gradient**.
    - Without reporting **peak VRAM**, **number of trainable parameters**, and **on-disk/host memory** for the anchor, the “comparable GPU-time” claim can be misleading.
    - A fair comparison should include **(i) peak VRAM, (ii) wall-clock time, (iii) GPU-hours, (iv) $S/S_D/S_G$** on **identical hardware**, and should also compare to **memory-efficient full-FT** methods (e.g., GaLore/Natural-GaLore, MeZO/Sparse-MeZO) rather than only PEFT.
- **Anchor-gradient justification lacks direct baselines.** The paper motivates anchoring via variance reduction and stability, but **does not directly compare** against established *gradient conflict/scale control* methods such as **PCGrad** or **GradNorm** under the same setups. Such comparisons are needed to validate that the *anchor-mixing* specifically is the right mechanism (vs. gradient surgery or reweighting).
- **“Finetune once” vs. hyperparameter sensitivity.** The reported sensitivity peak around a specific **k₀** (e.g., ~200/T) undercuts the “no retuning” narrative unless the authors provide a **robust heuristic** that generalizes across models/domains and include **mean±std** over multiple seeds.
- **Metric opacity risk.** Because $S$ is a harmonic mean over **normalized** scores, it can hide **absolute** degradations. Raw (unnormalized) scores and **Δ from the base model** should be reported alongside $S$.
- **Dependency on “general data.”** The approach **requires** access to a sufficiently representative “general” dataset to estimate the anchor. Reproducibility, licensing, and **sensitivity to the composition/size** of this dataset are not fully characterized.

**Questions:**

1. **Fair-cost table.** For Direct/Vanilla FT, LoRA/DoRA, GaLore (and variants), and DBA, please report on **the same hardware**: **peak VRAM, wall-clock time, GPU-hours, number of trainable parameters, on-disk/host memory for the anchor**, and final **$S, S_D, S_G$**.
2. **Anchor vs. gradient-surgery baselines.** Under the same experimental setups (including the News-QA OOD split), how does DBA compare against **PCGrad** and **GradNorm** in terms of $S/S_D/S_G$, early-stage stability (loss/grad-norm curves), and convergence speed?
3. **k₀ robustness.** Does the **k₀** optimum generalize across domains/models/seeds? Please provide **multi-seed sweeps** (mean±std) and, if applicable, a **default heuristic** that is demonstrably robust.
4. **General-data sensitivity.** How does performance vary when the **general dataset** used to estimate the anchor changes in **composition** or **size**? Please report variance across at least **three** distinct general datasets.
5. **Raw metrics.** Alongside $S$, please provide **raw, unnormalized** scores (and **Δ from the base model**) for general benchmarks (e.g., MMLU, MMLU-Pro, GSM8K, MATH, M3Exam) and domain metrics to ensure the harmonic mean is not masking regressions.
6. **Anchor overhead.** What is the **memory and IO overhead per step** from storing/restoring the rank-truncated anchor gradient during Stage-2 training, and how does it affect **throughput**?

---

> ### Author Response · Authors · 2025-11-20
> **Responses to the Reviewer UAVW [1/2]**
>
> Thank you for your detailed review. We appreciate you highlighting our method's ability to **"mitigate catastrophic forgetting"** and achieve a **"performance balance"**, and for noting the **"practically appealing"** nature of its reusable global signal.
>
> W1 & Q1: Fairness of Cost Comparison
>
> Firstly, DBA does not introduce additional memory or I/O overhead for the storage and repeated use of low-rank anchor gradients. The sharded low-rank anchor gradients are pre-loaded once into each GPU's VRAM prior to training, ensuring all operations are conducted entirely on the GPU.
>
> We further present the training overhead of DBA and Direct Fine-tuning (FT) for Llama3.1-8B on the News QA dataset using 16 A100 GPUs. As shown in the table below, DBA's low-rank gradient anchors occupy 5.5 GB of memory. Relative to Direct Fine-tuning (FT), DBA exhibits a modest 0.7 GB increase in Peak VRAM, due to the sharded gradient anchors and the computational cost for gradient correction. Owing to the fact that the anchor gradients are loaded only once, the wall-clock time increased slightly. We will include the training overhead and the detailed analysis in the Appendix of the revised manuscript.
>
> method | anchor memory | Peak VRAM | trained parameters | walk-clock time | GPU Hour |  $S$
> Direct FT | - | 37.1GB | 7B | 3.91h | 3.40h | 50.44
> DBA | 5.5GB | 37.8GB | 7B | 4.20h | 3.45h | 63.76
>
> Secondly, Table 3 in our paper includes PEFT methods such as LoRA and Galore (along with their respective GPU times). However, DBA's primary objective is not to reduce fine-tuning costs. Instead, its main comparison with these methods focuses on their efficacy in mitigating catastrophic forgetting, rather than their training overhead.
>
> Lastly, as metioned in Review aLnJ' Q1, DBA is not in conflict with PEFT methods. Specifically, DBA can be applied within the low-rank space of LoRA to further mitigate catastrophic forgetting. This will be explored in our future work.
>
> W2 & Q2: Comparison with Other Methods
>
> This is a valuable question. Firstly, DBA and gradient surgery method (PCGrad and GradNorm) exhibit distinct differences. The objective of DBA is to mitigate catastrophic forgetting of general domain knowledge during LLMs fine-tuning, rather than multi-task learning on general and specific tasks. Conversely, PCGrad is employed in multi-task reinforcement learning to suppress inter-task interference by orthogonally projecting task gradients. GradNorm, on the other hand, balance task gradients through the adaptive adjustment of sub-task loss weights in multi-task learning.
>
> Secondly, PCGrad and GradNorm require the individual storage of gradients for each task. This leads to excessive GPU memory requirements, which makes these methods impractical for direct application in full-parameter fine-tuning of LLMs. We anticipate that leveraging similar methodologies for catastrophic forgetting mitigation could demonstrate significant potential. We will further investigate this direction in our future work.
>
> W3 & Q3: "Finetune Once" and Hyperparameter $k_0$
>
> We appreciate the comment regarding the clarity of "Finetune Once." To be more precise, our claim posits that a singular fine-tuning process on the target domain data is adequate, thereby obviating the necessity for iterative experimentation driven by different data mixing strategies.
>
> To further assess the robustness of our results to variations in $k_0$, we extended the sensitivity analysis presented in Figure 6 of Appendix B.2. As shown in the table below, for $k_0$ values ranging from $150/T$ to $300/T$, the impact on $S$ remains within 2%."
>
> | $k_0$ | $S$ |
> |:---:|:---:|
> | $150/T$ | 61.98 |
> | $175/T$ | 62.92 |
> | $200/T$ | **63.76** |
> | $225/T$ | 63.28 |
> | $250/T$ | 63.50 |
> | $275/T$ | 63.12 |
> | $300/T$ | 63.39 |
>
> Moreover, we propose the heuristic $k_0 \propto 1/T$ (where $T$ denotes the total training steps) as a practical guideline. This relationship is effective because it maintains a consistent total contribution from global gradients irrespective of the training duration.
>
> W4 & Q5: Metric Opacity and Raw Scores
>
> Agree. It appears that there may be a misunderstanding. To clarify, the use of 'normalized' in section 4.3 signifies that all raw scores for the evaluated benchmarks were within the interval of 0 to 1. Crucially, no external normalization procedures were applied during the computation of $S_D$ , $S_G$ , and $S$. We will revise the wording in the paper to ensure this distinction is clear.
>
> While the averaged $S_G$ score was designed to facilitate overall performance on general domain, we acknowledge that raw scores are equally important. Due to page limitations, these raw scores were not included in the submitted version. Due to character limitations, we now provide the raw scores for experiments on News QA domain here. In the revised version of the paper, we will add a new section in the Appendix, detailing the raw scores of all models on general benchmarks.

---

> ### Author Response · Authors · 2025-11-20
> **Responses to the Reviewer UAVW [2/2]**
>
> | Metric | Direct FT | Vanilla FT | LoRA | DoRA | Galore | DBA (Ours) |
> |---|:---:|:---:|:---:|:---:|:---:|:---:|
> | **Llama3.1** | | | | | | |
> | $T \downarrow$ | 1.27 | 14.45 | **0.94** | $\underline{0.99}$ | 2.02 | 1.32 |
> | $S_D \uparrow$ | **82.38** | 80.62 | 71.23 | 73.71 | 79.13 | $\underline{82.37}$ |
> | $S_{MMLU}$ | 30.50 | 34.62 | 37.79 | 38.80 | **42.95** | $\underline{40.84}$ |
> | $S_{MMLU-Pro}$ | 34.28 | 37.22 | 45.95 | **48.29** | $\underline{46.20}$ | 44.16 |
> | $S_{GSM8k}$ | 35.56 | 39.19 | 48.14 | 50.23 | $\underline{53.13}$ | **54.54** |
> | $S_{MATH}$ | 40.76 | 42.13 | 50.48 | $\underline{58.30}$ | **58.51** | 56.67 |
> | $S_{M3Exam}$ | 40.60 | 45.48 | 57.84 | 56.99 | **64.45** | $\underline{63.80}$ |
> | $S_G \uparrow$ | 36.34 | 39.73 | 48.04 | 50.52 | **53.05** | $\underline{52.00}$ |
> | $S \uparrow$ | 50.44 | 53.22 | 57.38 | 59.95 | $\underline{63.52}$ | **63.76** |
> | **Phi4** | | | | | | |
> | $T \downarrow$ | 2.32 | 16.51 | **1.72** | $\underline{1.80}$ | 3.70 | 2.42 |
> | $S_D \uparrow$ | 87.38 | $\underline{88.23}$ | 83.12 | 83.37 | 85.02 | **89.19** |
> | $S_{MMLU}$ | 20.21 | 59.90 | **62.67** | $\underline{62.47}$ | 60.55 | 59.08 |
> | $S_{MMLU-Pro}$ | 25.35 | 59.66 | 66.96 | $\underline{68.79}$ | 66.79 | **71.97** |
> | $S_{GSM8k}$ | 27.52 | 71.83 | **81.24** | 73.19 | 74.46 | $\underline{77.04}$ |
> | $S_{MATH}$ | 27.33 | 80.72 | 84.03 | 80.52 | **89.89** | $\underline{85.05}$ |
> | $S_{M3Exam}$ | 31.59 | 88.44 | 92.40 | **98.51** | 89.66 | $\underline{96.41}$ |
> | $S_G \uparrow$ | 26.40 | 72.11 | $\underline{77.46}$ | 76.68 | 76.27 | **77.91** |
> | $S \uparrow$ | 40.55 | 79.36 | 80.19 | 79.88 | $\underline{80.41}$ | **83.17** |
> | **Qwen3** | | | | | | |
> | $T \downarrow$ | 1.27 | 14.45 | **0.93** | $\underline{0.99}$ | 2.03 | 1.28 |
> | $S_D \uparrow$ | 79.37 | 80.27 | 70.27 | 70.78 | **80.88** | $\underline{80.82}$ |
> | $S_{MMLU}$ | 3.06 | 42.19 | **55.48** | **55.48** | $\underline{51.70}$ | 51.32 |
> | $S_{MMLU-Pro}$ | 3.54 | 43.08 | 57.80 | $\underline{64.78}$ | 59.68 | **65.66** |
> | $S_{GSM8k}$ | 3.95 | 51.63 | $\underline{66.11}$ | 65.15 | 66.43 | **67.81** |
> | $S_{MATH}$ | 4.24 | 60.06 | $\underline{76.27}$ | 72.94 | **83.20** | 76.73 |
> | $S_{M3Exam}$ | 4.36 | 64.84 | **81.90** | 80.69 | $\underline{81.14}$ | 81.08 |
> | $S_G \uparrow$ | 3.83 | 52.36 | 67.51 | 67.81 | $\underline{68.43}$ | **68.52** |
> | $S \uparrow$ | 7.31 | 63.38 | 68.86 | 69.26 | $\underline{74.14}$ | **74.16** |
>
> W5 & Q4: Sensitivity to General Data
>
> The general dataset used in our paper was filtered from a raw pool of approximately 2.4M samples using the Platypus [1] method. To evaluate sensitivity, we sampled various ratios from this 2.4M raw pool. As shown in the table below, performance ($S$) remains remarkably stable as data volume increases, and even drops slightly at 100% usage. This aligns with findings from LIMA [2], suggesting that data quality and curation strategy outweigh sheer quantity. This confirms that $\hat{g}_G$ is robust and does not require massive amounts of raw SFT data to be effective.
>
> | Sample Ratio (of 2.4M) | **20% (Used in paper)** | 40% | 60% | 80% | 100% |
> | :--- | :---: | :---: | :---: | :---: | :---: |
> | $S$ | **63.76** | 63.55 | 63.68 | 63.71 | 63.12 |
>
> Furthermore, deriving $\hat{g}_G$ from pre-training or post-pretraining data yields even better results, as these gradients align more closely with the base model's intrinsic distribution. We conducted a preliminary experiment where we curated 100B tokens of post-pretraining data to compute $\hat{g}_G$, comparing it against the anchor derived from standard SFT data used in our main paper. As shown in the table below, the anchor derived from post-pretraining data achieved a superior joint performance score ($S$) on the News QA task.
>
> | Anchor Source | $S_D$ | $S_G$ | $S$ |
> | :--- | :---: | :---: | :---: |
> | General SFT Data (Ours) | 82.37 | 52.00 | 63.76 |
> | **Post-Pretrain Data** | **82.51** | **53.85** | **65.17** |
>
> Consequently, we propose that generating $\hat{g}_G$ should ideally be the responsibility of upstream developers. We view $\hat{g}_G$ as a standard "Model Artifact" as critical as model weights. We advocate for pre-training developers to release these gradients alongside their models. This allows end-users to simply download and load the artifact, completely eliminating the need for data collection, sensitivity tuning, or copyright concerns on the user side.
>
> [1] Lee, A., et al. "Platypus: Quick, cheap, and powerful refinement of llms." arXiv preprint arXiv:2308.07317 (2023).
> [2] Zhou, C., et al. "Lima: Less is more for alignment." NeurIPS (2024).
>
>
> Q6: The cost of graident anchor
>
> As clarified in response to W1 & Q1, DBA does not introduce additional memory or I/O overhead for the storage and repeated use of low-rank anchor gradients. The sharded low-rank anchor gradients are pre-loaded once into each GPU's VRAM prior to training, ensuring all operations are conducted entirely on the GPU.

---

> > ### Author Response · Authors · 2025-11-26
> > **Gentle Reminder regarding our Rebuttal**
> >
> > Dear Reviewer UAVW,
> >
> > We hope this message finds you well. As the discussion period is drawing to a close, we respectfully invite you to review our response and the revised manuscript.
> >
> > In our rebuttal, we have specifically addressed your concerns regarding cost fairness by providing a detailed breakdown of Peak VRAM, wall-clock time, and anchor memory usage (demonstrating minimal overhead). We also clarified the justification for the anchor gradient and provided additional sensitivity analyses for the hyperparameter $k_0$.
> >
> > We value your assessment and hope our response has adequately addressed your concerns. We look forward to your feedback.
> >
> > Best regards,
> >
> > The Authors

---

### Author Response · Authors · 2025-12-01
**Summary of Rebuttal**

We sincerely thank the Area Chair and the Reviewers for their time and effort in handling our paper. We are grateful for the constructive feedback, which has significantly improved the quality of our work. We are encouraged by T8Kj's decision to raise the score to 6 following our clarifications. The other three reviewers were unable to respond in time because of the “API Security Incident”. However, we believe our detailed responses and revisions have effectively addressed their concerns.

**Strengths:**
We are encouraged that the reviewers unanimously agree our work addresses a **"timely and practically relevant"** problem (**PtTU, aLnJ, T8Kj**), recognizing the proposed method as an **"innovative"** (**PtTU**) and **"intuitive"** (**T8Kj**) decoupling strategy. Specifically, the reviewers highlighted:

*   **Efficiency & Practicality:** (**PtTU and aLnJ**) commended the method as a "highly practical and efficient alternative to data mixing," noting it tackles a "costly problem" with a "massive advantage" in reducing GPU hours.
*   **Performance Balance:** (**UAVW**) noted that DAA effectively mitigates catastrophic forgetting, achieving a "superior harmonic mean score" and maintaining a strong balance between domain and general capabilities.
*   **Reusability:** (**UAVW**) found the global gradient signal to be "practically appealing," as it allows for efficient reuse across different domains.

Besides, we have fully addressed their initial concerns in our revised manuscript and detailed responses. Below is a summary of the key improvements:

**Methodological Clarification & Renaming (Addressing aLnJ, PtTU & T8Kj)**
*   **Renaming:** Incorporating **Reviewer aLnJ**'s suggestion, we renamed the method from **DBA** to **Dynamic Anchor Annealing (DAA)** to avoid terminological confusion with Gradient Boosting.
*   **Theoretical Grounding:** Addressing **Reviewer PtTU**, we revised **Section 3.1** to ground our Global Gradient Anchoring (GGA) in **Variance Reduction** (Eq. 4) and **Trajectory Smoothing** (Fig. 2), replacing the previous SVRG analogy with a more robust theoretical justification.
*   **Differentiation:** We clarified the distinction between our optimization-level decoupling and Multi-Task Learning (**Reviewer T8Kj**) in **Section 2.1**.

**Enhanced Experimental Rigor (Addressing T8Kj & UAVW)**
*   **Significance Testing:** To address **Reviewer T8Kj**'s concern regarding performance margins, we added Statistical Significance Tests on the News QA benchmark (**Appendix G**), confirming our improvements are statistically significant ($p < 0.001$).
*   **Transparency:** We provided Raw Experimental Scores for all general benchmarks (**Appendix J**) to address **Reviewer UAVW**'s concern about metric opacity.

**Comprehensive Cost & Sensitivity Analysis (Addressing UAVW)**
*   **Fair Cost Comparison:** We added a detailed breakdown of Peak VRAM, Wall-clock time, and Anchor memory usage (**Appendix H**), demonstrating that DAA incurs minimal overhead compared to direct fine-tuning.
*   **Robustness:** We included sensitivity analyses for the hyperparameter $k_0$, general data quantity, and anchor sources (**Appendix I**), demonstrating the method's stability.

**Future Directions (Addressing aLnJ)**
*   **PEFT Compatibility:** As suggested by **Reviewer aLnJ**, we added a discussion in **Section 5** on the compatibility of DAA with Parameter-Efficient Fine-Tuning (PEFT) methods, outlining a promising path for combining computational and memory efficiency.

**Conclusion:**
In summary, **Dynamic Anchor Annealing (DAA)** offers a streamlined solution to the "Efficiency-Performance-Generalization" trilemma in domain fine-tuning. By decoupling general and domain learning via a global gradient anchor, DAA eliminates the need for expensive data mixture searches, reducing GPU hours by **~91%** while achieving superior joint performance. We believe this work provides a valuable, scalable paradigm for the community.

We have $\color{blue}{highlighted\ all\ modifications}$ in the revised manuscript.

---

### Note · Authors · 2026-03-05

I have read and agree with the venue's withdrawal policy on behalf of myself and my co-authors.

---

### Meta-Review · Area_Chair_wkEm · 2026-01-06

**Summary:**

This paper proposes Dynamic Anchor Annealing (DAA) (formerly DBA), a full-parameter fine-tuning framework that aims to mitigate catastrophic forgetting during domain adaptation of LLMs without relying on data mixture. The key idea is to pre-compute a global gradient anchor from general data and use it to regularize domain fine-tuning through gradient anchoring, dynamic correction based on gradient similarity, and annealed learning rates. Empirical results across multiple models and domains show improved balance between domain performance and general capability retention, measured via a harmonic mean metric, and substantial savings in GPU hours compared to repeated data-mixture searches.

Reviewers generally agree that the problem is timely and practically relevant, and that the proposed method is intuitive and well implemented. However, concerns remain regarding conceptual positioning, theoretical grounding, and the scope of comparisons, which together prevent a clear consensus that the paper meets the ICLR acceptance bar at this time.

**Reviewer Concerns:**

**Concerns largely addressed in the rebuttal:**

- **Terminology and framing confusion** *(aLnJ W1; T8Kj W1)*
  Reviewers noted that the original terminology (e.g., “Gradient Boosted,” “decoupling”) was misleading and suggested architectural backbone–head separation. The authors addressed this by renaming the method (DBA → Dynamic Anchor Annealing / Global Gradient Anchoring) and clarifying that the approach operates at the optimization level within full-parameter fine-tuning.

- **Result robustness and statistical significance** *(T8Kj W3)*
  The absence of significance testing was raised as a concern. The authors added statistical significance tests (e.g., on News QA), which Reviewer T8Kj explicitly stated resolved their concerns and led them to raise their score.

- **Metric transparency and cost reporting** *(UAVW W1, W4)*
  Reviewers requested clearer reporting of raw scores and cost metrics. The rebuttal provided raw benchmark results, clarified the construction of the harmonic-mean metric S, and added a detailed breakdown of peak VRAM, wall-clock time, and anchor memory usage.

**Remaining limitations:**

- **Scope mismatch between continual-learning motivation and evaluated baselines**
  *(T8Kj W1; PtTU W3)*
  While the paper motivates the method using catastrophic forgetting and continual learning language, empirical comparisons are largely restricted to full fine-tuning and PEFT baselines. The relationship to established continual learning or gradient-conflict mitigation methods remains mostly conceptual rather than empirically demonstrated.

- **Workflow dependence of efficiency claims**
  *(PtTU W3; UAVW W1)*
  The reported ~91% GPU-hour savings compare DAA against the full process of searching over multiple data-mixture ratios, rather than a single optimally tuned vanilla fine-tuning run. While the authors clarify that eliminating this search is the intended advantage, the generality of the efficiency gains depends on whether such workflows reflect standard practice.

- **Hyperparameter robustness (largely addressed)**
  *(UAVW W3)*
  The rebuttal provides additional sensitivity analysis showing that performance is stable over a reasonably wide range of k0 values on a per-task basis. While demonstrating full task-agnostic optimality of a single k0 setting would be ideal, the authors fixed k0 using a simple heuristic (k0 ∝ 1/T) and reused the same value across all tasks and models in their experiments, without per-task tuning. This addresses the primary concern about brittleness in practice, and the remaining limitation is largely about ideal generality rather than practical soundness.

**Reviewer Scores:**

- **Reviewer T8Kj**: Explicitly stated that their concerns were satisfactorily addressed in the rebuttal and raised their score from 4 to 6.

- **Reviewer PtTU**: Indicated that their assessment shifted from a negative stance to a neutral/borderline position after reading the rebuttal, and stated they would be comfortable with either accept or reject. While unable to update the score in the system, their comments suggest a slight upward adjustment (from 4 to borderline).

- **Reviewer UAVW**: Did not provide a post-rebuttal update. However, most of their concrete concerns (cost reporting, metric transparency, and k0 robustness) were substantively addressed in the rebuttal. Given their original comment that they “would not mind if the paper is accepted,” it is plausible that their score would increase slightly, e.g., from 4 to around 5.

- **Reviewer aLnJ**: Maintained a positive assessment throughout (score 6) and did not raise new blocking concerns during discussion. No significant score change is expected.

Overall, based on explicit score updates and post-rebuttal comments, a reasonable post-discussion interpretation of the reviews is approximately 6, 6, 5, and 5, corresponding to a borderline accept profile.

---

### Decision · Program_Chairs · 2026-01-26

Reject